# Comparative Genomic Analysis Reveals the Emergence of ST-231 and ST-395 *Klebsiella pneumoniae* Strains Associated with the High Transmissibility of *bla_KPC_* Plasmids

**DOI:** 10.3390/microorganisms11102411

**Published:** 2023-09-27

**Authors:** Muna AL-Muzahmi, Meher Rizvi, Munawr AL-Quraini, Zakariya AL-Muharrmi, Zaaima AL-Jabri

**Affiliations:** 1Medical Laboratory, Diwan Health Centre, Muscat 130, Oman; m.muzahmi@gmail.com; 2Department of Microbiology and Immunology, College of Medicine and Health Sciences, Sultan Qaboos University, Muscat 123, Oman; rizvimeher@squ.edu.om; 3Microbiology and Immunology Diagnostic Laboratory, Department of Microbiology and Immunology, Sultan Qaboos University Hospital, Muscat 123, Oman; munawer@squ.edu.om (M.A.-Q.); muharrmi@squ.edu.om (Z.A.-M.)

**Keywords:** mobile genetic elements, anti-microbial resistance, *Klebsiella pneumoniae*, whole-genome sequencing, integrons, plasmids, sequence types, phylogenetic analysis

## Abstract

Conjugative transposons in Gram-negative bacteria have a significant role in the dissemination of antibiotic-resistance-conferring genes between bacteria. This study aims to genomically characterize plasmids and conjugative transposons carrying integrons in clinical isolates of *Klebsiella pneumoniae. The* genetic composition of conjugative transposons and phenotypic assessment of 50 multidrug-resistant *K. pneumoniae* isolates from a tertiary-care hospital (SQUH), Muscat, Oman, were investigated. Horizontal transferability was investigated by filter mating conjugation experiments. Whole-genome sequencing (WGS) was performed to determine the sequence type (ST), acquired resistome, and plasmidome of integron-carrying strains. Class 1 integrons were detected in 96% of isolates and, among integron-positive isolates, 18 stains contained variable regions. Horizontal transferability by conjugation confirmed the successful transfer of integrons between cells and WGS confirmed their presence in conjugative plasmids. Dihydrofolate reductase (*dfrA14*) was the most prevalent (34.8%) gene cassette in class 1 integrons. MLST analysis detected predominantly ST-231 and ST-395. *Bla*_OXA-232_ and *bla_CTX-M-15_* were the most frequently detected carbapenemases and beta-lactamases in the sequenced isolates. This study highlighted the high transmissibility of MDR-conferring conjugative plasmids in clinical isolates of *K. pneumoniae*. Therefore, the wise use of antibiotics and the adherence to effective infection control measures are necessary to limit the further dissemination of multidrug-resistant bacteria.

## 1. Introduction

*Klebsiella pneumoniae* is an opportunistic pathogen that is associated with different serious nosocomial infections, including pneumonia, septicemia, meningitis, and urinary tract infections (UTIs) [1,2]. To treat the infections caused by extended-spectrum beta-lactamase (ESBL)-producing *K. pneumoniae,* carbapenem antibiotics are the drugs of choice; however, they are considered as last-resort antibiotics [3]. Moreover, the number of *K. pneumoniae* carbapenemase (KPC) enzyme producers has been increasing [4,5,6,7]. The spread of resistance determinants has been facilitated by horizontal gene transfer mechanisms mainly via mobile genetic elements (MGEs). Integrons, for example, are semi-mobile platforms that recognize and capture mobile gene cassettes and transform them to usable genes by ensuring their appropriate expression [8,9,10]. They are located on different MGEs, such as plasmids, transposons, and pathogenicity islands (PAIs), which enable their movement between different bacteria [11,12]. A number of studies showed that most *K. pneumoniae* clinical isolates carried class 1 integrons, whereas class 2 integrons were only present in 1–2% of the isolates and rarely harbored class 3 integron [13]. A significant association between integron-positive isolates and antibiotic resistance for some drugs was observed, including piperacillin-tazobactam, ciprofloxacin, cefotaxime, and ceftazidime [12,14,15,16,17]. Both class 1 and class 2 integrons have gene cassettes encoding resistance to trimethoprim (*dfr*) as a predominant gene, which may be due to the long-term usage of this antibiotic [18,19,20].

Class 1 integron is associated with transposons derived from Tn*402*, which can bind in a larger transposon such as Tn*21*. Over 80 different antibiotic-resistance-encoding gene cassettes are associated with class 1 integrons, but most integrons have the gene *aadA*, which encodes streptomycin–spectinomycin resistance [8]. Moreover, class 1 integrons have a 3′-conserved segment that contains a *sulI* gene-encoding resistance to sulfonamides and a *qac*EΔ1 gene-encoding resistance to quaternary ammonium compounds [21]. In Oman, there has been an increasing concern regarding the spread of MDR *K. pneumoniae* and KPC producers in clinical settings [22,23,24]. However, molecular characterization is still lacking. Therefore, we aim, in this study, to genomically analyze MDR *K. pneumoniae* carrying class 1 integrons, in terms of the structure, sequence types (STs), and antimicrobial phenotype and genotype. In addition, we investigate the association between antimicrobial susceptibility and the presence of integrons in clinical isolates of *K. pneumoniae*. Finally, we conduct conjugation experiments to evaluate the horizontal-transfer capability, which is key to predict the future dissemination and implementation of intervention strategies.

## 2. Materials and Methods

### 2.1. Bacterial Strains

A total of fifty (*n* = 50: ESBL (*n* = 27), XDR (*n* = 21), and PDR (*n* = 2) strains = *K. pneumoniae* isolates were collected in this study from the Diagnostic Microbiology and Immunology Laboratory in Sultan Qaboos University Hospital (SQUH), Muscat, Sultanate of Oman between July 2019 and October 2019. The isolates were mostly from urine (*n* = 25), respiratory (*n* = 10), wound (*n* = 9), bloodstream (*n* = 4), body fluid (*n* = 1), and biopsy (*n* = 1) samples. All isolates were processed as per the Clinical and Laboratory Standards Institute guidelines [25]. The isolates were cultured in a selective medium (Cefuroxime Cysteine Lactose Electrolyte-Deficient agar, Oxoid, Basingstoke Hampshire, UK) prior to preservation in sterile CryoBeads (Mast Diagnostics, Derby, UK) at −80 °C for further analysis (Table 1).

### 2.2. Antimicrobial Susceptibility

The antimicrobial susceptibility profiles of the *K. pneumoniae* isolates were assessed using two methods: disk diffusion method and BD Phoenix^TM^ automated system (Becton Dickinson Diagnostic Systems, Sparks, MD, USA). For both methods, three to five bacterial colonies from overnight pure cultures were suspended in normal saline (Fisher Chemical, Cramlington, UK) and adjusted to a 0.5 McFarland standard (approximately 1–2 × 10^8^ CFU/mL) using a CrystalSpec nephelometer (BD Diagnostics, Sparks Glencoe, MD, USA), according to the manufacturer’s recommendations. After 15 min, the suspension was spread onto a Mueller–Hinton agar (MHA) surface (Oxoid, Hampshire, UK) and left for 1–2 min at room temperature to be absorbed. After 15 min, the selected antibiotic disks (BioMérieux and Liofilchem, Germany) were placed on the inoculated MHA plates using sterile forceps. The plates were incubated at 37 °C for 18–24 h. The antibiotics were selected according to the CLSI standard as follows: ampicillin (AMP 10 µg), piperacillin/tazobactam (TZP 110 µg), cefepime (FEP 30 µg), cefotaxime (CTX 30 µg), cefoxitin (FOX 30 µg), ceftazidime (CAZ 30 µg), imipenem (IPM 10 µg), meropenem (MEM 10 µg), amikacin (AK 30 µg), gentamicin (CN 10 µg), and ciprofloxacin (CIP 5 µg). *E. coli* (ATCC 25922) and *P. aeruginosa* (ATCC 27853) were used as the susceptible control strains. The interpretive categories and zone diameter breakpoints, nearest to the whole mm for each antibiotic, are listed in Table 2. The BD Phoenix^TM^ automated system was used to test the susceptibility of colistin and tigecycline, and to confirm the antimicrobial susceptibility of other antibiotics tested by the disk diffusion method. For colistin, >1 mg/L was considered resistant according to CLSI breakpoints [25]. The European Committee on Antimicrobial Susceptibility Testing (EUCAST) interpretive criteria was used for tigecycline, with ≤0.5 mg/L considered as sensitive [26]. The ESBL production was further confirmed using the disk diffusion method. Both cefotaxime and ceftazidime alone and in combination with clavulanic acid were used. An increase in diameter of ≥5 mm with the clavulanic acid compared to the antibiotic alone was interpreted as positive for ESBL production. The XDR and PDR isolates were characterized genotypically to the carbapenemase genotype level using Xpert Carba-R (Cepheid, Sunnyvale, CA, USA).

### 2.3. Genomic DNA Extraction and Purification

The DNA was extracted using Qiagen kit (QIAamp^®^ genomic DNA kit, Hilden, Germany) as described in the manufacturer’s instructions with slight modifications. One to four colonies were suspended in 10 mL Mueller–Hinton broth (Oxoid, Mampshire, UK) and left overnight on a shaking incubator set to 250 rpm at 37 °C. Then, the bacterial suspension was centrifuged for 15 min at 4000× *g*. The amount of 20–40 mg of pelleted bacterial cells was re-suspended in a pre-lysis buffer (100 μL TE buffer and 0.1 μL RNase A 100 mg/mL) (Thermo Fisher Scientific, Winsford, UK). The cells were well resuspended by pipetting up and down several times. After that, the samples were incubated again at 37 °C at a 400 rpm shaking incubator for 30–60 min (Innova 4000, New Brunswick Scientific, Hertfordshire, UK). A volume of 1μL Proteinase K (stock concentration = 20 mg/mL) + 99 μL TE were added to the sample to obtain a final volume of 200 μL in 1.5 mL microcentrifuge tube. Then, 200 μL of the sample was mixed with 400 μL of lysis solution and incubated at 65 °C for 15 min using a heat block (Eppendorf ThermoStat plus, Hamburg, Germany). As per the manufacturer’s instruction, the samples were washed twice using the columns. The DNA pellet was eluted in a final volume of 50–100 μL of nuclease-free H_2_O (QIAamp^®^ genomic DNA kit, Hilden, Germany). The extracted DNA was aliquoted into 2 vials, which were stored at 4 °C and −80 °C for future use. The boiling method was used to extract the DNA from trans-conjugant colonies in the conjugation experiment [28]. Three to five colonies were removed from the plate and resuspended in 30 μL of nuclease-free water. Then, the samples were heated at 100 °C for 10 min in a heat block. Cells were then pelleted by centrifuging the samples at 8000× *g* for 1 min. The supernatant was used as a template for PCR.

### 2.4. Polymerase Chain Reaction (PCR)

#### 2.4.1. Detection and Characterization of Classes 1, 2, and 3 Integrons

PCR assays were performed using Go-*Taq* DNA polymerase (Promega Ltd., Madison, WI, USA). The cycle conditions were adjusted depending on the gene size and the primer’s melting temperature. The detection of classes 1, 2, and 3 integrons was investigated by the amplification of integrase genes *intl1, intl2*, and *intl3,* respectively. The PCR reaction mixture consisted of 5 μL of 5X PCR green buffer; 0.5 mL of dNTPs (10 μM); 1 μL of both forward and reverse primers (10 μM); 0.1 μL of Go-Tag DNA polymerase (5 μ/mL); and 1.5 μL of template DNA. Then, nuclease-free water was added until the total volume reached 25 μL. A reaction mixture without a DNA template was used as a negative control. The same PCR reaction mixture was used to amplify variable-region genes. For the DNA that was extracted by the boiling method, dimethyl sulphoxide (DMSO) was added to encourage the annealing of the primers to the template and further enhance the amplification. A thermocycler (Eppendorf master cycler^®^, Merk, Darmstadt, Germany) was used for PCR amplification and the cycling conditions for Go-Tag enzyme are mentioned in Table 3. Annealing temperatures, which are different for each gene depending on the primer’s melting temperature, are listed in Table 4. The specific primers for detecting integrase genes and integron-specific variable regions were used as previously described [18]. Gel electrophoresis results can be found in the Appendix A.

#### 2.4.2. PCR Purification

PCR products were purified using QIAquick PCR and Gel Cleanup Kit (Qiagen, Hilden, Germany). Five volumes of buffer PB were added to 1 volume of the PCR sample and then mixed. To bind DNA, the mixture was applied to the QIAquick column and centrifuged for 30–60 s at 17,900× *g* at room temperature. After centrifugation, the column was placed in a clean 2 mL collection tube and the tube containing the filtrate was discarded. Then, 750 μL of buffer PE was added to the QIAquick column and centrifuged again for 30–60 s at the same previous conditions. Again, the column was placed in a clean 2 mL collection tube and the tube containing the filtrate was discarded. Then, the QIAquick column was centrifuged once more for 1 min to remove the residual wash buffer. Finally, the QIAquick column was placed in a clean 1.5 mL microcentrifuge tube to which 30 mL of elution buffer (10 mM Tris·HCl, pH 8.5) was added. The tube was incubated at room temperature for 1 min and then centrifuged for 30–60 s at 17,900× *g*. The purified DNA was stored at −80 °C for future use.

### 2.5. Gel Electrophoresis

All PCR products were visualized by using 2% agarose gel electrophoresis containing 1× TBE (ThermoFisher Scientific, Waltham, MA, USA) and MIDORI Green Direct (NIPPON-genetics, Europe). The bands were visualized by using G:Box Chemi-XR5 device (Syngene, India). DNA fragments sizes were estimated by using the GeneRuler 1 kb DNA Ladder (ThermoFisher Scientific, USA).

### 2.6. Whole-Genome DNA Sequencing

Whole-genome sequencing (WGS) in this study was performed at microbesNG by Illumina next-generation sequencing at a minimum coverage of 30× (https://microbesng.co.uk/, Birmingham, accessed on 20 July 2022). The genomic DNA was prepared and then was sent for sequencing following the protocol provided by the sequencing facility. The assembled sequences were then retrieved from the websites and analyzed accordingly.

### 2.7. Bioinformatics Analyses

WGS data of *K. pneumoniae* isolates files were analyzed using the center for genomic epidemiology server (CGE) (http://cge.cbs.dtu.dk/services, accessed on 2 July 2022). At CGE, the CSI phylogeny tool [29] was used to investigate the relatedness between the strains based on single nucleotide polymorphism (SNP) identification. The assembled genome data for all samples were returned to the server as input. The reference genome was *K. pneumoniae* subsp. *pneumoniae* HS11286 (GeneBank: CP003200). The SNPs were localized and filtered based on the default settings, which include a minimum distance of ten bases between the SNPs, a sequencing depth of ten bases, and a minimum SNPs quality of 30. In addition, the Z-score for each SNP must be above 1.96. Then, the data file was visualized and managed by using an online tool, Interactive Tree of Life (iTOL) [30]. The ResFinder tool from the GCE server was used in this study to detect the acquired antimicrobial resistance genes and their specific location on the sequence [31]. MLST and PlasmidFinder were used to detect Multilocus sequence typing (MLST) and plasmids, respectively [32,33]. Along with ResFinder tool in CGE, the Comprehensive Antibiotic Resistance Database (CARD) (https://card.mcmaster.ca/home, accessed on 20 July 2020) was used to detect the presence of putative antibiotic-resistance genes by using the Resistance gene identifier (RGI) tool. This tool, in addition, can detect known point mutations within the resistance-conferring genes. Artemis software version (18.2.0) from the Sanger Institute was used for the visualization of WGS features. Moreover, GenBank from the National Center for Biotechnology Information (NCBI) (https://www.ncbi.nlm.nih.gov/genbank/, accessed on 3 August 2020) was used to search in the DNA database for similar DNA sequences and the Basic Local Alignment Search Tool (BLAST) was used to search for similarities between DNA sequences (https://blast.ncbi.nlm.nih.gov/Blast.cgi, accessed on 3 August 2020).

### 2.8. Conjugation Experiment

Conjugation experiments were performed for *K. pneumoniae* strains with variable regions as donors, and *E. coli* HB101 as a recipient by using the filter mating method [34]. To obtain the standard concentration of the required antibiotic (Amoxicillin and streptomycin (Sigma-Aldrich, St. Louis, MO, USA)) solution, the powder was weighed on a calibrated analytical balance and dissolved in an appropriate solvent (as per the manufacturer’s instructions). *K. pneumoniae* isolates were cultured in a Luria agar (LA) (Lab M^®^, Manchester, UK) plate with 100 mg/mL of amoxicillin. The *E. coli* HB101 strain was cultured in the Luria agar (LA) plate with 50 mg/mL of streptomycin. Both strains were incubated overnight at 37 °C. On the next day, a single colony from previous plates was inoculated in 5 mL of Luria broth (LB) (Lab M^®^, Manchester, UK) with the same antibiotic selection and incubated overnight at 37 °C on a shaking incubator. Then, a 1:100 dilution of both cultures was inoculated in 5 mL fresh LB and incubated again on the shaking incubator at 37 °C until the bacteria grew to their exponential phase (OD_600_ 0.2), which was measured using Bio-Photometer (Eppendorf, Germany). After that, 1 mL of each culture was centrifuged at 8000× *g* for 5 min to harvest the cells. The pellets were washed twice with 1X Phosphate-Buffered Saline (PBS) (prepared from 10X PBS, pH 7.4, ThermoFisher Scientific, MA, USA) to remove the remaining antibiotics. Then, the pellets were re-suspended in 200 µL of PBS. The recipient, donor, and mixed bacterial suspension were spotted onto the center of the 0.45 mm pore size membrane (Whatman Ltd. Sigma-Aldrich, St. Louis, MO, USA) and placed on LA without antibiotics. The spots were allowed to dry and then incubated overnight at 37 °C. Subsequently, filter papers were washed in 1 mL 1X PBS to detach the bacterial cells. Serial dilutions up to 10–8 were performed for the donor, recipient, and mixed bacteria by using PBS as a diluent. The trans-conjugants are selected on an LA plate supplemented with both amoxicillin and streptomycin. For colony-forming unit (CFU) counts, dilutions of the donor were plated on LA with 100 µg/mL of amoxicillin and the recipient in 50 µg/mL of streptomycin. All plates were incubated at 37 °C overnight. The efficiency of trans-conjugation is calculated using the following formulas:% Efficiency of transconjugation = nTransconjugants/nRecipient Recovered (Cfu/mL) × 100

## 3. Results

Class 1 integrons were detected in 92% of isolates (46/50) using IntI1F and IntI1R primers (Table 4). The screening of classes 2 and 3 integrons by detecting the presence of *IntI2* and *IntI3* genes, respectively, showed that these were not present in the collection. The analysis of the antimicrobial susceptibility patterns of the 50 *K. pneumoniae* isolates was performed (Figure 1). All clinical isolates were resistant to ampicillin (100%); most of the isolates were resistant to cefotaxime (98%), cefepime (92%) and ceftazidime (84%); and almost half of them showed resistance to piperacillin/tazobactam (48%). Moreover, 88% of the isolates showed resistance to ciprofloxacin. Furthermore, these isolates showed intermediate resistance to gentamicin (54%), imipenem (52%), and, to a lesser extent, to amikacin and colistin (30% and 6%).

According to the results obtained by the PhoenixBD semi-automated system, 54% of *K. pneumoniae* isolates (*n* = 27) were identified as ESBLs, whereas 42% (*n* = 21) of isolates were identified as XDR and two isolates (4%) as PDR. Almost all XDRs were susceptible to colistin, except three isolates (Kp 2, Kp 22, and Kp 50) (Table 5). In addition, 11.8% and 5.6% of the XDRs were susceptible to amikacin and gentamicin, respectively. Some of the XDR isolates were susceptible to amikacin, gentamicin, imipenem, and meropenem along with colistin (Table 5).

The conjugation was performed between variable region-positive isolates as donors and *E. coli* HB101 strain as a recipient. Seven representative strains were selected for testing (Kp 5, Kp 21, Kp 27, Kp 37, Kp 42, Kp 49, and Kp 50). The trans-conjugant PCR products are shown in (Appendix A). All isolates were positive for the *IntI1* gene, which confirmed the successful occurrence of conjugation via horizontal transfer. In the tested strains, the constituents of gene cassettes, plasmids, and sequencing types were analyzed by sequencing. The WGS data show that, in our tested strains, plasmids were detected in all *E. coli* trans-conjugants, which confirms that these conjugative plasmids are highly transmissible in the *E. coli* conjugation experiments. The efficiency of trans-conjugation for each strain was calculated and ranged between 0.07% and 17.3% (Figure 2). This exemplifies the role of these plasmids as the main vehicle for the transmission of integrons, and subsequently the transport of various types of gene cassettes from the donor to the recipient strains.

Out of the total 50 isolates of *K. pneumoniae*, 24 isolates were analyzed by WGS, including the 7 isolates which were successful in conjugation. Sixteen of these isolates were positive for the variable regions (Kp 5, 7, 10, 11, 15, 16, 21, 28, 37, 40, 42, 43, 44, 45, 46, and 49) and two isolates showed bands with an unexpected size for the *intI3* gene (Kp 4 and Kp 22). According to previous studies, these bands could be variants of integrase genes [35]. Among the 24 isolates, 3 were integron-negative isolates by the PCR method (Kp 4, Kp 27, and Kp 30). However, WGS showed that two of these isolates were positive for class 1 integron (Kp 27 and Kp 30), whereas Kp 4 was still negative. Therefore, Kp 4 isolate was excluded from the subsequent molecular analysis.

The MLST of all sequenced isolates was determined by the online MLST database from the Center of Genomics Epidemiology (CGE) [32]. Nine of the isolates belong to sequence type (ST-231) (Kp 5, 6, 7, 10, 11, 15, 28, 30, and 45), six isolates are ST-395 (Kp16, 22, 41, 43, 44, and 50), and two isolates are ST-405 (Kp 42 and Kp 46). Only one isolate belongs to each of the following miscellaneous sequence types: ST-37 (Kp 40), ST-45 (Kp 21), ST-147 (Kp 49), ST-280 (Kp 27), ST-1710 (Kp37), and ST-1741(Kp 25).

The WGS data of 23 integron-positive isolates were used to construct a whole-genome phylogenetic single-nucleotide polymorphism (SNP) study (Figure 3), and the output from the analysis shows three main clusters of strains. The first cluster is composed of nine isolates belonging to ST-231. The second cluster of strains is in the ST-395 group (*n* = 6). The clinical data of the above-mentioned two STs indicate the frequent moving of patients from the Intensive Care Unit (ICU) to the male medical ward or vice versa. However, the third cluster consisted of two strains only belonging to ST-405. The remaining strains belong to miscellaneous sequence types (ST-37, ST-147, ST-280, ST-1741, ST-45, ST-13, ST-17, and ST-1710).

A close-up analysis of the two predominant STs (ST-231 and ST-395) was conducted to predict if there are any possible recent outbreaks through clustering (Figure 4). The SNP tree for ST-231 and ST-395 was created individually and it was observed that the PDR isolates (Kp6 and Kp11) branched from the same clade, suggesting a possible vertical transmission among patients. Similarly, among the other isolates, ST-231 and ST-395 stains are very closely related.

The demographic data for thirteen strains that belong to ST-231 (*n* = 9) and ST-395 (*n* = 4), which account for most strains, are listed in Table 6. Ninety-two percent of the patients were males aged over 50 years (69.2%). The length of stay varied between patients, ranging from 1 to 134 days. Most of the patients were admitted to the ICU (61.5%) and the rest of the patients were admitted to intermediate care wards (male medical wards). Highly critical ICU patients usually require broad-spectrum antibiotics, which exert tremendous selection pressure, thus driving antimicrobial-resistance bacteria to thrive further in these settings.

At the level of ST, more than two-thirds (77.8%) of ST-231 and all ST-395 strains were isolated from patients admitted either to the ICU or male medical wards or moved between the two places interchangeably. Almost all patients have various risk factors contributing to their likelihood of acquiring highly resistant *K. pneumoniae* strains, including ICU admission, ventilation, urinary catheters, central venous catheters, and hospitalization, for more than 7 days [36]. Unfortunately, the mortality rate among these patients was very high (84.6%).

All sequenced isolates carried a wide range of acquired antimicrobial-resistance-conferring genes. Fourteen isolates harbored the *bla_OXA-232_* gene, all of which were XDR and PDR isolates except KP 7, which was the only XDR isolate that did not carry the *bla_OXA-232_* gene (Figure 3). *bla-_NDM5_* was detected in only one isolate (Kp 49). Based on the WGS data and Carba-R, 87% of XDR and PDR isolates (20/23) have *bla_OXA-48-like_*, 8.7% (2/23) have *bla_NDM_*, and one isolate (4.3%) has both *bla_NDM_* and *bla_OXA48-like_* genes. There was a high level of agreement between the WGS and GeneXpert results for the tested isolates (8/9). The *bla_OXA-1_* allele was detected in eight isolates, most of which belong to ST-395 (*n* = 6). The ESBL gene *bla_CTX-M-15_* was detected in all isolates, while *bla_TEM-1B_* was seen in 15 isolates. Moreover, 10 isolates were positive for *bla_SHV-1_*, 9 isolates for *bla_SHV-11_*, 2 for *bla_SHV-76_*, and 2 for *bla_SHV-27_*.

A number of various genes conferring resistance against aminoglycosides were detected, including *aac(6′)-Ib-cr, aac(6′)-Ib Hangzhou*, *aadA1,2*, *aac(3)-lId*, *aph(3′)-Ia, strA, strB, armA*, and *rmtB.* A fosfomycin-resistance-conferring gene (*fosA6*) was detected in all isolates except Kp 49, which harbored *fosA5*. In addition, six different genes encoding quinolone resistance were detected in the isolates, which are *aac(6′)-Ib-cr*, *oqxA, oqxB*, *qnrS1*, *B1*, and *B66*. Moreover, rifampicin- *(arr-2)* and trimethoprim-resistance-encoding genes (*dfr*) were found in 10 and 11 isolates, respectively.

The tetracycline-resistance gene, *tetA*, was detected in four ESBL isolates, whereas *tetD* in two XDR isolates. According to the Phoenix automated system, all of these were tigecycline-resistant, except one ESBL isolate that carried the *tetA* gene (Kp 46). The gene of the *acrAB* efflux pumps’ regulator (*marA*), which contributes to resistance against tigecycline and other antibiotic classes, was detected in all sequenced isolates. In addition, *K. pneumoniae* efflux pumps (Kpn E, F, G, and H) that confer resistance to different antibiotic classes, such as macrolide, aminoglycoside, cephalosporin, tetracycline, rifamycin, and colistin, were detected in all isolates.

The four colistin-resistance isolates belonging to ST-231 and ST-395 (Kp 6, Kp 11, Kp 22, and Kp 50) were negative for plasmid-mediated *mcr* genes; therefore, a further analysis of the SNPs in PhoPQ operons was performed. No chromosomal mutations were detected in the regulatory two-component systems (TCSs) PmrAB and crrAB, and in the *mgrB* gene, a negative regulator gene of TCSs. However, all isolates harbored the regulatory TCSs (PhoPQ). Our analysis of the PhoPQ operons in the colistin-resistant isolates identified four novel, undescribed SNPs in the PhoP genes with the amino acid substitutions as follows: (Val130Glu), (Gln147His), (Gln131Glu), and (Pro129Leu) (Table 7). The significance of these SNPs is yet to be determined, as none of these substitutions have been described.

Most of the tested isolates showed an agreement between their resistance phenotypic patterns to different antibiotic families and the presence of resistance genes, as shown in Table 8. Carbapenems, cephalosporin, and quinolone antibiotics showed a complete level of agreement between phenotype and genotype. However, aminoglycoside antibiotics (AK, CN) showed a low level of agreement, with 21 isolates carrying resistance genes and only 9 and 15 isolates with phenotypic-resistance patterns to amikacin and gentamicin, respectively. This observation could be because these resistance-conferring genes might not be expressed. The expression of these genes needs to be further investigated, which was not within the scope of this study.

Using WGS data, the variable region of 23 isolates, including the trans-conjugant strains (Kp 5, Kp 21, Kp 27, Kp 37, Kp 42, Kp 49, and Kp 50), were analyzed to detect gene cassettes (Table 9 and Figure 3). Ten different gene cassettes were identified in 16 strains with a variable region, including those encoding resistance to aminoglycoside (*aadA*, *aacA4*, *aac(6′)-Ib, APH(3″)-Ia,* and *ant1*) trimethoprim (*dfrA5*, *dfrA12*, and *dfrA14*), rifampin (*arr2* and *arr3*), chloramphenicol (*catB3, catB8*, and *cmlA1*), macrolide (*ereA2* and *erm*), and quaternary ammonium compound *(emrE).* The most frequent gene cassette was composed of the *dfrA14* gene alone, which was present in eight isolates (34.8%). However, three isolates showed empty integrons (In0) without any gene cassette insertions (Kp 16, Kp 22, and Kp 25). One of them had a variable region (Kp 16), while the other two did not (Kp 22 and Kp 25).

The comparative analysis of integrons was performed by locating the gene cassettes within the variable region. Mapping was conducted based on the genes between the primers of the variable region that varied among strains (Table 9). It was observed that there was a contig break in the middle of the integron cassettes in some strains and the downstream sequence (3′-end) was found in a separate contig. Therefore, the genetic maps of the integrons were drawn after the manual reassembly of the various components (Figure 5).

Among the detected plasmids, eight of them harbored antibiotic-resistance genes (Figure 6 and Table 10). The most prevalent plasmid is pKPQIL-IT, which is present in 15 strains that belong to ST-395 and ST-231. It is a 115,300 bp in size, IncFIB(QIL) replicon carrying genes associated with resistance to β-lactams (*bla_TEM-1_* and *bla_KPC-3_*).

Plasmid pKP3-A is a 7605 bp linear ColKP3 replicon, carrying the *bla_OXA-181_* gene. ST-231 and ST-405 share the uniqueness of pKP3-A that the integron is interrupted by insertion sequences (Figure 5). These isolates exhibit highly similar genotypes. All produced OXA-181, and the majority also have plasmid mediated-*bla_TEM-1_*, *bla_KPC-3_* genes. Unlike Xpert Carba-R, the plasmid finder tool detected the *kpc* gene from WGS data in all ST-231 and ST-405 isolates. This could suggest that the copy numbers of the plasmid might be low, or the gene might have not been expressed and hence could not be detected by the GeneXpert system.

Isolates Kp 41 and Kp 50 are the only two strains carrying IncFIB (pNDM-Mar) and IncHI1B (pNDM-Mar) plasmids. In these isolates, *bla_NDM-1_* was within the pNDM-Mar plasmid. Isolate Kp 49 (ST-147) had *bla_NDM-1_* as part of the (ble-bla_NMD-1_) operon, where it was flanked by bleomycin-resistance (*ble*) gene and N-(5′-phosphoribosyl) anthranilate isomerase (*trpF*) gene. Moreover, the plasmid composition of Kp 49 had IncR, IncFII, and pKP3-A. The plasmid pAMA1167-NDM-5 is present only in seven of the ST-231 isolates. It is a 11,310 bp sized, IncFII(pAMA1167-NDM-5) replicon with genes encoding resistance against aminoglycosides (*aph(3″)-Ib*, *aph(6)-Id, aadA2*, *aadA5*, *aac(3)-IIa*, and *aac(6*′*)-Ib-cr5*), β-lactams (*bla*_NDM-5_, *bla*_OXA-1_, *bla*_CTX-M-15-1_, and *bla*_TEM-1_), chloramphenicol (*cat)*, sulfonamides (*Sul1* and *Sul2*), trimethoprim (*dfrA12* and *dfrA17*), tetracycline (*tet(b)* and *tet(c)*)*,* and macrolides *(emrE* and *mp(A)*). The sequence analysis of the pKPN-IT plasmid revealed that it is a 208,191-bp IncFIB(K) replicon carrying (*aadA2*, *cat*, *Mph(A)*, *Sul1*, and *dfrA12*) genes. It is found in the KP 10, 25, 27, 37, 42, and 46 isolates. However, the plasmid pBK30683 is a 139,941-bp FIA replicon present only in one isolate (KP 37).

## 4. Discussion

The present study focused on the molecular characterizations of integrons as a common class of mobile elements and their significance in the dissemination of multi-drug resistance genes among *K. pneumoniae*. Whole-genome sequence (WGS) data were analyzed to determine the sequence type as well as characterization of integron-carrying plasmids. In addition, WGS was used for studying the correlation between antimicrobial-resistance genotype and phenotype, and the level of agreement was analyzed. Moreover, the phylogenetic relatedness of the isolates was associated with the patient’s demographic data to explore the possible spread of mobile elements. Most of the studies in the region investigate the epidemiology of AMR, with a few studies concerning the prevalence of mobile genetic elements, such as genomic islands and their molecular characteristics, in Gram-negative bacteria (GNB), such as *A. baumannii* and *K. pneumoniae* [22,23,37,38,39,40,41]. Moreover, another study that was conducted in Gram-negative bacteria isolates in Palestinian hospitals focused on integrons and their role in AMR dissemination [42]. To our knowledge, this is the first study in Oman that assesses the transmissibility of plasmids by examining the efficiency of conjugation in vitro, thus emphasizing the importance of this mechanism in the spontaneous transfer of integron-carried plasmids intra-specially.

In the current study, 96% (48/50) of our *K. pneumoniae* isolates carried class 1 integrons, whereas none of the isolates contained either classes 2 or class 3 integrons. This finding is similar to the results of previous studies that were conducted in Iran [43,44]. In addition, a previous study showed that 100% of the MDR isolates were found to be positive for class 1 integrons, 36% for class 2, and none for class 3 integrons [12]. However, other studies detected class 2 integrons at low prevalence levels of 1.7% and 8.3% [42,45].

A high prevalence of class 1 integrons among our MDR isolates was observed and therefore it was the focus of this study. The presence of class 1 integrons in most of the isolates suggests that these genetic elements confer a significant advantage to their hosts, where exposure to antibiotic overuse creates a selective pressure in hospital environments [12,46]. The absence of both classes 2 and class 3 integrons from the isolates in this study is expected since these classes are relatively uncommon [8,47,48,49]. Moreover, this might be the case because our selection is limited to representative MDR isolates, which are unusually difficult to treat over a short period of time, which might not reflect the actual prevalence. Therefore, a larger sample size, including screening specimens, is more likely to reveal the other classes of integrons.

Our findings show that 37.5% (18/48) of integron-positive isolates were carrying variable regions. By the analysis of WGS data, a total of 10 different gene cassettes were detected in these isolates. The latter isolates encode for aminoglycoside, trimethoprim, rifampin, chloramphenicol, macrolide, and quaternary ammonium compounds. Trimethoprim-resistant gene cassettes (*dfrA14*), which encodes for dihydrofolate reductase enzymes, were found to be predominant in the isolates in this study (34.7%). Several studies showed that *dfr* genes were the most frequent gene cassettes present in integrons [18,42,50,51]. The stability of this gene cassette in class 1 integrons might indicate the overuse of the trimethoprim for a long period of time, specially to treat urinary tract infections [18]. Moreover, despite the initial results of the variable region, which was detected by employing conventional PCR using primers that bind in the 5′- and 3′-conserved regions, further WGS analysis showed that 13% of isolates with positive variable regions carried empty gene cassettes. Moreover, previous studies reported the presence of *sul1* at the 3′CS region of the integrons [52,53,54]. However, it was only observed in (3/20) isolates, despite the amplification of the 5′CS–3′CS region by PCR. This has been described in *Salmonella enterica* [55,56]. The non-classical structure of integrons, where *intI1* and/or 5′ end of integrons is truncated, were also observed in two strains (Kp 16 and Kp 25), which might suggest that amelioration of *IntI1* confers a selective advantage for the host bacteria [35].

Simple PCR assays tend to overestimate the presence of integron cassettes and cannot provide further information on the subsequent changes in the integron cassettes, such as deletions and insertions. In fact, gene cassettes undergo continuous changes in their compositions, which means that these cassettes can integrate antimicrobial-resistant genes once conditions are favorable. Truncations and inversions within the integron structures are not uncommon and could not be further detected by simple PCR. Therefore, WGS data were key in bridging the gap on the various components within these integrons.

Plasmids are considered as main carriers for antibiotic-resistant genes through HGT mechanisms. Spontaneous plasmid conjugation is possible when the plasmid has a compatible mechanism of transfer. A plasmid is classified as conjugative when it helps other conjugative elements, such as integrons, to move between different strains [57]. In the current study, the trans-conjugant colonies were positive for *intI1* gene in all the representative strains, with trans-conjugation efficiency ranging between 0.05% and 17.3%. This confirmed the successful transfer of integrons between cells via HGT. WGS data for the tested isolates supports our hypothesis, as plasmids were identified in all the tested isolates, which is similar to the result of a previous study [18]. Being carried on plasmids, these integrons might have a significant fitness cost since more copies of plasmids can be present compared to chromosomally located integrons. This observation has been reported in previous studies on *K. pneumoniae* as well as other Gram-negative organisms [58,59].

Most of our sequenced isolates showed a hierarchical relationship between different types of MGEs. The detected gene cassettes were embedded inside integrons, which were in turn inserted inside transposons, which are flanked by insertion sequences (IS) that can confer mobility to transposons. Some of these composite transposons were incorporated into conjugative plasmids that offer transfer to other cells [60]. Most of the isolates in this study (Kp 5, 21, 27, 28, 37, 40, 41, 42, 44, 46, 49, and Kp 50) have integrons inserted in transposons that are flanked by IS (Figure 4). All above-mentioned strains had a similar IS, which is IS*6* and IS*6100* (Kp 42 and Kp 46). IS*6100* belongs to the IS*6* family of transposable elements forming co-integrates as an endpoint of transposition, which was originally isolated from *Mycobacterium fortuitum* [61]. In general, different previous studies showed the same relationship, where the integrons were in transposons that were imbedded in a conjugative plasmid [62,63]. However, MGEs, including Tns and ISs, disrupt the integron structure.

The presence of IS*6* on both sides of the Dcm-methylation operon, which is a type II restriction–modification (RM) system and next to the integron denotes that RM systems, are mobile and located in plasmids. These systems are involved in genome rearrangements and enhance virulence and resistance plasmid dissemination by carriage on other MGEs, as in our study on integrons [64,65,66]. The existence of these IS*6* in the isolates in this study entails their importance in disseminating antibiotic-resistance genes among various genera and species of bacteria [67,68]. In this study, the specific capture of resistance cassettes by the integrons signifies apparent bias driven by the selective pressure of antibiotic therapy regimes. Other studies have also shown that integrons could carry catabolic genes, which proves that integrons play a broader role in bacterial evolution [10,69].

ST-231, the most abundant STs found in this study, is widely distributed in South-West Asia with clonal dissemination in Singapore, Brunei, and Darussalam between 2013 and 2015 [70,71]. In addition, in India, ST-231 strains were reported as a predominant ST in 22 isolates (45%) [72]. In Europe, the first occurrence of the MDR *K. pneumoniae* ST-231 clone was confirmed in Switzerland [73]. This dissemination may represent a global public threat toward a new epidemic clone. ST-395 has been reported as the most common MDR *K. pneumoniae* clone (69%) in North-Eastern France as well as an outbreak in an ICU in Italy [74,75].

In a previous study conducted in isolates from the Arabian Peninsula, different ST types were detected in MDR *K. pneumoniae * in Gulf countries, including Oman; however, ST-231 was not among these STs [76]. Similarly, previous studies were conducted in Oman and Saudi Arabia that have concluded that no ST-231 strain was found [77,78]. The absence of ST-231 possibly indicates a recent emergence of ST-231 in Oman. However, ST-147 was one of the commonly detected STs from *K. pneumoniae* isolates in the two previous studies [76,77], whereas in the present study, only one strain belongs to ST-147.

Both of these STs, ST-395 carrying strains and 77.8% of ST-231, were clustered in the same wards (mainly medical and ICU wards) with a significant mortality rate (84.6%). This observation highly suggests that there is a horizontal transfer of resistance-conferring genes, which is alarming. The heavy use of broad-spectrum antibiotics for critically ill patients in high-dependency areas and ICU patients selects for highly resistant strains and enhances the spread of resistance determinants. Furthermore, this may indicate the inefficient infection control practices that allowed the dissemination of plasmid-mediated resistance in the hospital. This finding corresponds to a study that was conducted in carbapenem-resistant GNB in SQUH, where 87% of infections were healthcare-associated with a 62% mortality rate [23].

Over the last 15 to 20 years, carbapenem-hydrolyzing β-lactamases, including OXA-48-like and NDM-type carbapenemases, disseminated and emerged in Enterobacteriaceae all over the world [79,80]. Carbapenemase OXA-48-like differs from the classical OXA-48 by one to five amino acids, in which they hydrolyze carbapenems and penicillins but do not affect extended-spectrum cephalosporin [80].

OXA-232-producing *K. pneumoniae* (OXA232Kp) was identified for the first time on a 6.1-kb ColE-type non-conjugative plasmid in France in 2013 from a patient who returned from India and, since then, it has spread worldwide [79,80,81]. OXA-232 is considered as a point-mutation derivative from OXA-181 with one amino acid difference [82]. In this study, WGS and Xpert Carba-R data detected the presence of the *bla*_OXA-48-like_ gene in 87% (20/23) of our XDR and PDR isolates. These genes were identified by WGS as *bla*_OXA-232_ in 93.3% (14/15) of our XDR- and PDR-sequenced isolates. All of them belonged to ST-231 and ST-395. The level of agreement between WGS- and GeneXpert-tested isolates was highly significant (*n* = 8/9), with only one isolate that was negative for the OXA gene at GeneXpert and positive in WGS (Kp 43). Illumina WGS has good coverage for sequencing the whole genome, whereas GeneXpert is based on real-time PCR, and even with 97% specificity, it might miss some allelic variants. The plasmid analysis of our strains showed that the OXA-181 gene (or OXA-232) was carried on pKP3-A (7605 bp) replicon-type ColKP3. It was expressed only in the XDR- and PDR OXA-232-positive strains. This is similar to a previous study, where 33% and 100% of OXA-232 isolates were carried on ColKP3 plasmid in the USA and China, respectively [82,83].

On the other hand, NDM was first detected in 2008 in a Swedish patient of Indian origin and then spread all over the world [84]. In this study, Xpert Carba-R results show that only 4% (2/50) of the isolates have the NDM gene, whereas only one isolate (4.3%) that belongs to ST-147 harbors both OXA-48-like (*bla*_OXA-232_) and NDM genes (*bla*_NDM-5_) according to the WGS data. A study was conducted in the UAE to characterize carbapenem-resistant Enterobacteriaceae in the Arabian Peninsula, showing that NDM and OXA-48-like genes are the most-detected genes with rates of 46.5% and 32.5%, respectively. In the isolates collected from Oman, 46% and 44.4% of the isolates were carrying NDM and OXA-48-like genes (*bla*_OXA-181_), respectively, whereas only 1.6% carried both genes [76].

Along with the carbapenem genes, ESBL genes, including *bla*(CTX-M), *bla*(TEM), and *bla*(SHV), were detected, and reported in different studies [71,72]. All *bla*(CTX-M) genes were identified, including *bla*_CTX-M-15_, and it was the dominant ESBL amongst our strains since it was produced by all sequenced isolates. One main factor that contributes to the CTX-M-type ESBL distribution in *K. pneumoniae* is conjugative plasmids, particularly those belonging to IncF with specific insertion sequences [85]. Moreover, all bla(TEM) genes belonged to bla_TEM-1_ and were detected in 69.6%, whereas SHV types were characterized as *bla_SHV-1_, bla_SHV-11_, bla_SHV-27_, and bla*_SHV-76_. All ST-231 strains carried *bla_SHV-1_*, while ST-395 strains carried *bla_SHV-11_*. Moreover, OXA-1 was positive in all ST-395 strains only with an association with other ESBL genes *(bla_SHV-11_, bla_CTX-M-15_, bla_TEM-1_)*. The association between OXA-1 and CTX-M-15 genes renders isolates resistant to β-lactam–β-lactamase inhibitor combinations. Noteworthily, a wide variety of GNB carried the OXA-1 gene in plasmid and integron locations [86]. Plasmid analysis in this study determined that the *bla_OXA-1_* gene along with *bla*_CTX-M-15_ was located in IncF plasmids IncFII and IncFII (pAMA1167-NDM-5).

Resistance to colistin, a drug that is used as the last line drug in the treatment of extensively resistant pathogens, is increasingly reported in Enterobacteriaceae, particularly *K. pneumoniae* [87]. Several mechanisms are involved in colistin resistance, mostly the emergence of mobilized colistin-resistance (*mcr*) genes via plasmid and the mutations in the chromosomal gene (*mgrB*) and operons (PmrAB and PhoPQ), which are associated with the biosynthesis and modification of lipopolysaccharide (LPS). The PhoPQ regulatory system is activated at low concentrations of Mg^2+^ or Ca^2+^ and acidic PH [88]. The PhoQ (sensor kinase) activates PhoP (regulator protein) by phosphorylation, which in turn activates *pmrFHIJKLM* operons. These operons lead to LPS modification by adding 4-amino-4-deoxy-L-arabinose (L-Ara4N) and phosphoethanolamine (PETN) to lipid A. This modification neutralizes the negative charge of LPS leading to the low affinity of LPS to positively charged colistin [88,89,90].

Four sequenced isolates were resistant to colistin via BMD; however, all of these strains were negative for the plasmid-mediated colistin-resistance gene *(mcr)* and harbored a truncated *mgrB* gene. Bioinformatics analysis revealed the presence of four SNPs in the PhoP genes with the following amino acid substitutions: (Val77Glu), (Gln147His), (Gln131Glu), and (Pro129Thr). A further analysis of the promoter region of PhoPQ operon did not reveal any SNPs. A previous study reported different SNPs in PhoP (Val3Phe and Ser86Leu) and PhoQ (Leu26Pro) that were found to have a role in colistin resistance when tested by real-time PCR [91]. However, the detected SNPs in this study have not been described before, and therefore, a further expression analysis by real-time PCR is needed to show their role along with the truncated *mgrB* gene in colistin resistance. Moreover, colistin resistance might occur due to other resistance mechanisms, such as capsule overproduction, which causes a reduction in the interaction between colistin and its target site at *K. pneumoniae* (LPS) [88,92]. Furthermore, all isolates in this study carried *K. pneumoniae* efflux pumps (Kpn E, F, G, H), which might contribute to colistin resistance.

Tigecycline has broad-spectrum activity against Gram-positive and Gram-negative bacteria and demonstrates efficacy and safety as salvage therapy for MDR/XDR bacteria [37,40,41,93,94]. However, tigecycline resistance is becoming more common in ESBL-producing, MDR, XDR, and carbapenem-resistant isolates [95,96]. In this study, most of the sequenced isolates (17/19) were tigecycline-resistant, while the remaining four isolates were not tested. The WGS data show that the efflux pumps’ regulator gene (*marA*) is present in all isolates (*n* = 23), including the two sensitive ones (Kp37 and Kp46). In this context, a study reported that the *marA* gene, along with other regulatory pathways, could mediate the resistance to tigecycline through upregulating the *acrAB* efflux pump, even in the absence of *ram A* [97]. In another study, the development of tigecycline resistance was attributed to the *tetA* gene [98]. In our study, the *tetA* gene was detected in four ESBL isolates, from which only two were resistant. Thus, it is more likely that the *marA* regulator is engaged in tigecycline resistance.

Several studies reported that the *armA* gene was in conjugative plasmids of the IncL/M or IncFIA plasmid, while *rmtB* in IncFI or IncFIA [99,100]. Interestingly, these two genes were only identified in Kp 49, which belongs to ST-147, and it is an OXA-232- and NDM-5-producing isolate. In addition, IncFIA(HI1) and IncFII plasmids were uniquely present in this isolate only, and a further search of *rmtB* and *armA* genes by BLAST using WGS data confirmed the localization of these two genes in the (IncFII) plasmid.

Among carbapenems, cephalosporin, and quinolones, all isolates possessed expressed genes with a complete agreement between AMR genotypic findings and phenotypic expression. These results are close to what has been detected in a previous study [101]. In contrast, aminoglycoside antibiotics (amikacin and gentamicin) had high non-expressed genes (57.1% and 28.6%, respectively). Since there are other mechanisms behind AMR, the presence or absence of certain genes might not be a sufficient indicator of the isolate’s resistance profile. Furthermore, genotypic analysis has to be correlated with the phenotypic findings of various mechanisms involved.

The versatility and ubiquity of integrons in bacterial genomes indicate the key role of these mobile elements in bacterial adaptation. The abundance of repeat sequences of ISs and Tns within the integron structures in our WGS data pertains to the tremendous selective pressure of the antibiotic regimens used in the hospital microbial environment. These integrons are very potent capture systems with a limitless capacity to exchange antibiotic gene cassettes as well as other genes that increase their fitness. Integrons with combinations of various antibiotic cassettes have been termed multi-resistance integrons (MRIs) in Gram-negative bacteria [102].

Genomic plasticity was apparent in a few cases where there was a deletion of non-essential genes or empty integrons, which confirms that these integron systems are performing adaptive rather than housekeeping functions. In support with this hypothesis, two of the isolates we examined had integrons with no antibiotic gene cassettes and the *intI1* gene was partially truncated (Kp 16 and Kp25). The different attachment sites and the variety of open reading frames of hypothetical or unknown functions within these integron systems are compelling. Further detailed analysis is required to unravel the importance of integrons among other MGEs towards the adaptation of bacterial evolution.

## 5. Conclusions

In summary, this study demonstrated a high prevalence of class 1 integrons (96%) in MDR *K. pneumoniae* in SQUH and the absence of classes 2 and class 3 integrons. The data of WGS confirmed the presence of diverse integron-carried gene cassettes, with *dfrA* as the predominant cassette. In addition, two main STs (ST-231 and ST-395) were detected with the dominance of OXA-232 carbapenemase, while the NDM-5 type was identified in one isolate. The sequence typing and genotypic characterization of the isolates by WGS revealed a possible break in the infection control in the male medical ward (R3) and ICU causing the intra-hospital transmission and spread of carbapenem resistance. The dissemination of integrons by the horizontal transfer of the conjugative plasmids play a vital role in the spread and exchange of resistance genes between bacteria of the same or different species [85,103]. This phenomenon imposes a tremendous threat to the currently available antibiotic regimens and necessitates the adoption of strict antimicrobial stewardship programs to prevent the further dissemination of AMROs as well as the improvement of infection control measures.

## Figures and Tables

**Figure 1 microorganisms-11-02411-f001:**
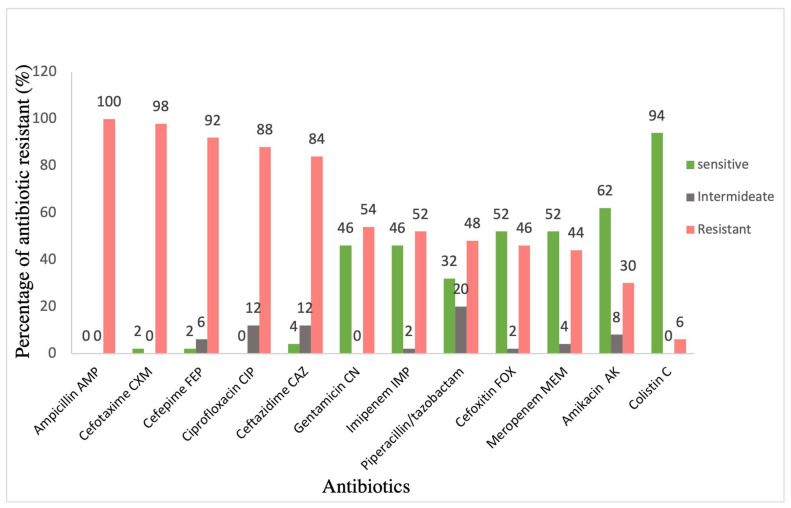
Bar chart representing the percentage of antimicrobial susceptibility of *K. pneumoniae* isolates against different antibiotics.

**Figure 2 microorganisms-11-02411-f002:**
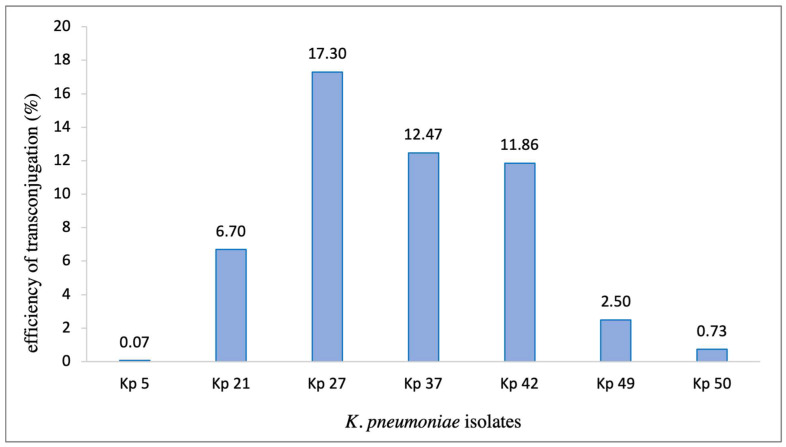
Bar chart representing the percentage of trans-conjugation efficiency of *K. pneumoniae* isolates as donors and *E. coli* HB101 as a recipient.

**Figure 3 microorganisms-11-02411-f003:**
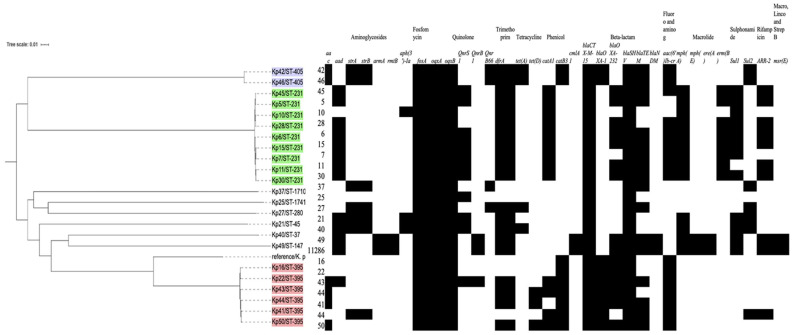
Maximum likelihood phylogenetic tree constructed from WGS data for 23 *K. pneumoniae* isolates. The phylogenetic tree is annotated with the isolate’s number and sequence type (ST) as follows: Green: ST-231, purple: ST-405, and red: ST-395. Black boxes to the right of each strain number illustrate the distribution of antibiotic-resistance genes; absent genes are shown in white boxes. *K. pneumoniae* HS11286 was used as a reference strain in this phylogenetic tree (accession number CP003200).

**Figure 4 microorganisms-11-02411-f004:**
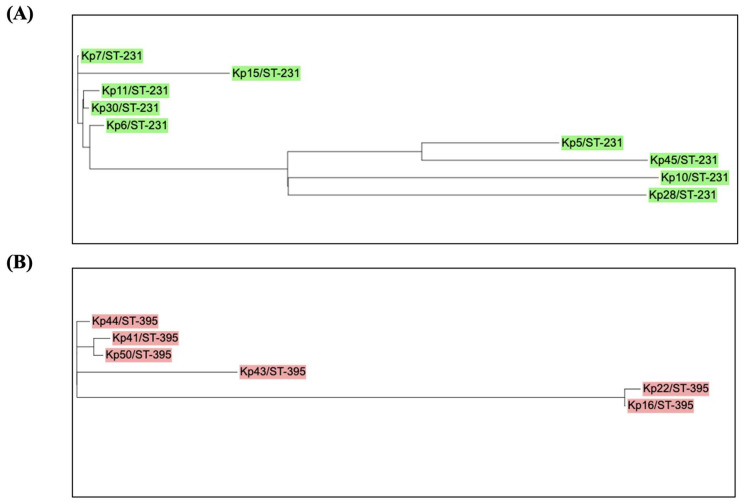
SNPs trees of *K. pneumoniae* strains (ST-231 and ST-395). A phylogeny tree showing the relatedness of the *K. pneumoniae* strains that belong to (**A**) ST-231 (*n* = 9) and (**B**) ST-395 (*n* = 6). Isolates are labeled according to their numbers and sequence type (ST). Green: ST-231 and red: ST-395. *K. pneumoniae* HS11286 was used as a reference isolate (accession number CP003200). An online tool (iTOL) was used to draw and edit the tree.

**Figure 5 microorganisms-11-02411-f005:**
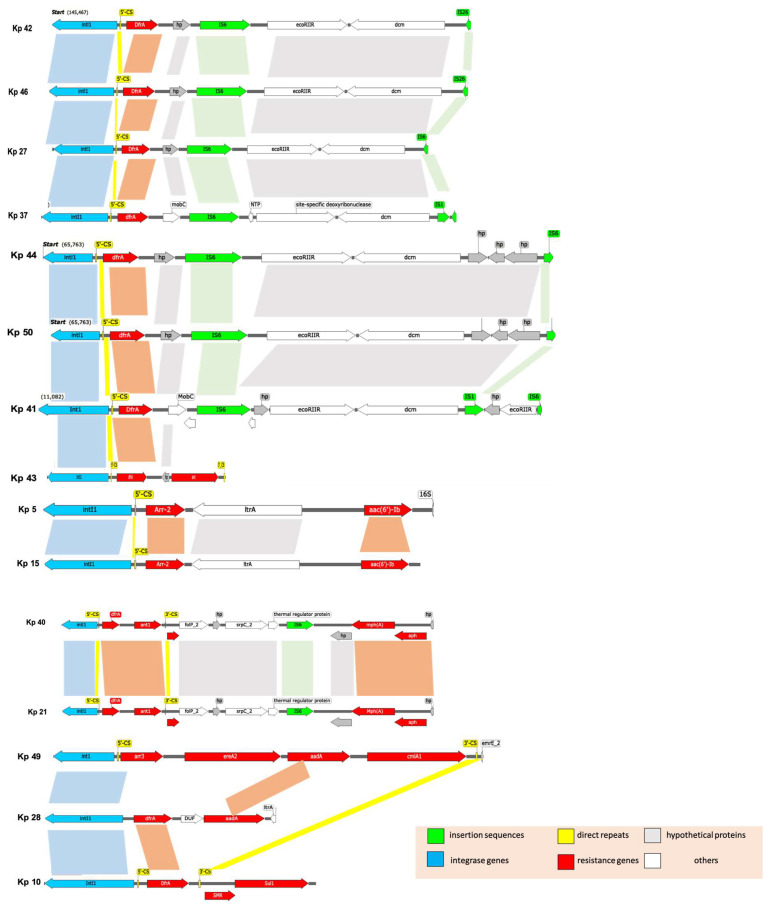
Schematic map of the composition of the integrons in *K. pneumoniae* strains. Genes and ORFs are denoted by arrowheads indicating the direction of transcription and colored based on the gene function classifications as shown in the key in the lower part of the figure. Shaded areas denote regions of homology (>95% nucleotide sequence identity). Grouping of strains is based on the sequence similarity as follows: ST-231 strains (Kp 41, Kp 43, Kp 44, and Kp 50), (Kp 27, Kp 37, Kp 42, and Kp 46), (Kp 40 and Kp 21), (Kp 5 and Kp 15) and (Kp 49, Kp 28, and Kp10).

**Figure 6 microorganisms-11-02411-f006:**
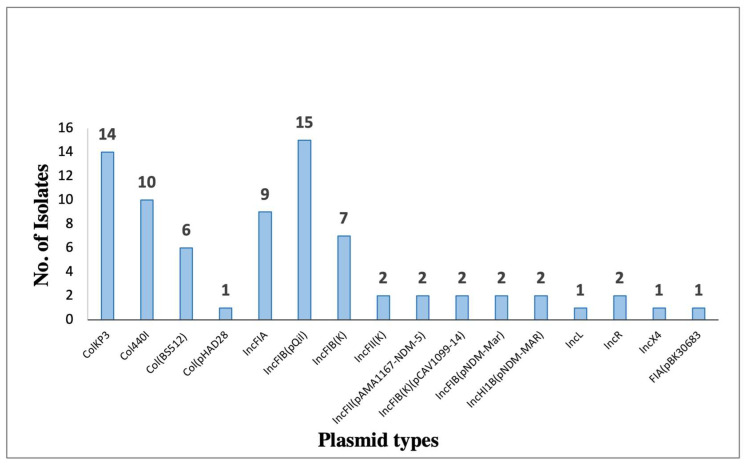
Bar chart representing the distribution of plasmid replicon types identified among *K. pneumoniae* (*n* = 23). Plasmid types were detected based on WGS data by using the ResFinder online tool.

**Table 1 microorganisms-11-02411-t001:** Clinical information of the *K. pneumoniae* isolates (*n* = 50).

Isolate	Month of Isolation	Type of Resistance	Specimen Type	Hospital Unit
Kp1	July	XDR	Tracheal aspirate	Emergency
Kp 2	July	XDR	Urine	Emergency
Kp 3	July	ESBL	Urine	Oncology
Kp 4	July	ESBL	Urine	Pediatrics
Kp 5	July	XDR	Urine	Male Medical
Kp 6	July	PAN	Tracheal aspirate	ICU
Kp 7	July	XDR	Wound	Male Medical
Kp 8	July	ESBL	Urine	Day Care
Kp 9	July	ESBL	Biopsy	Surgery
Kp 10	July	ESBL	Urine	Oncology
Kp 11	August	PAN	Tracheal aspirate	Male Medical
Kp 12	August	ESBL	Pus	Surgery
Kp 13	August	XDR	Blood culture	Emergency
Kp 14	August	ESBL	Wound	Male Medical
Kp 15	August	XDR	Catheter urine	Male Medical
Kp 16	August	XDR	Sputum	ICU
Kp 17	August	ESBL	Catheter urine	Pediatrics
Kp 18	August	ESBL	Urine	Emergency
Kp 19	August	ESBL	Urine	Pediatrics
Kp 20	August	ESBL	Urine	Pediatrics
Kp 21	August	ESBL	urine	Emergency
Kp 22	September	XDR	Wound	ICU
Kp 23	September	ESBL	Blood culture	Emergency
Kp 24	September	XDR	Skin	Emergency
Kp 25	September	ESBL	Peritoneal fluid	Male Medical
Kp 26	September	ESBL	wound	Surgery
Kp 27	September	ESBL	Urine	Emergency
Kp 28	September	XDR	Urine	Male Medical
Kp 29	September	XDR	Urine	Day Care
Kp 30	September	XDR	Wound	Male Medical
Kp 31	September	XDR	Sputum	ICU
Kp 32	September	XDR	bronchial wash	Emergency
Kp 33	September	ESBL	wound	Female Medical
Kp 34	September	ESBL	Urine	Pediatrics
Kp 35	September	ESBL	Blood culture	Female Medical
Kp 36	September	ESBL	Urine	Emergency
Kp 37	September	ESBL	Urine	Urology
Kp 38	October	ESBL	Urine	Emergency
Kp 39	October	ESBL	Urine	Emergency
Kp 40	October	ESBL	Tracheal aspirate	Neonatal unit
Kp 41	October	XDR	Wound	Male Medical
Kp 42	October	ESBL	Blood culture	Neonatal unit
Kp 43	October	XDR	Tracheal aspirate	Male Medical
Kp 44	October	XDR	Urine	Male Medical
Kp 45	October	XDR	Tracheal aspirate	Male Medical
Kp 46	October	ESBL	Tracheal aspirate	Neonatal unit
Kp 47	October	ESBL	Urine	Female Medical
Kp 48	October	XDR	Urine	Surgery
Kp 49	October	XDR	Urine	Male Medical
Kp 50	October	XDR	Urine	ICU

**Table 2 microorganisms-11-02411-t002:** Interpretive categories and zone diameter breakpoints, nearest to the whole mm [27].

		Zone Diameter Breakpoints (mm)		
Antibiotic	Disk Content	Susceptible	Intermediate	Resistant
Ampicillin	AMP 10 µg	≥17	14–16	≤13
Piperacillin–tazobactam	TZP 110 µg	≥21	18–20	≤17
Cefepime	FEP 30 µg	≥25	19–24	≤18
Cefotaxime	CTX 30 µg	≥26	23–25	≤22
Cefoxitin	FOX 30 µg	≥18	15–17	≤14
Ceftazidime	CAZ 30 µg	≥21	18–20	≤17
Imipenem	IMP 10 µg	≥23	20–22	≤19
Meropenem	MEM 10 µg	≥23	20–22	≤18
Gentamicin	CN 30 µg	≥15	13–14	≤12
Amikacin	AK 10 µg	≥17	15–16	≤14
Ciprofloxacin	CIP 5 µg	≥31	21–30	≤20

**Table 3 microorganisms-11-02411-t003:** PCR cycling conditions for Go-Taq polymerases.

Step	Temperature	Time
Initial denaturation	95 °C	2 min
Denaturation	95 °C	30 s
Annealing		30 s
Extension	72 °C	1 min/kb
Final extension	72 °C	10 min
Hold	15 °C	

**Table 4 microorganisms-11-02411-t004:** List of the specific primers used in this study. All primers were obtained from [18].

Gene		Annealing Temperature	Nucleotide Sequence (5′-3′)	Expected Size
Class 1 integrase gene	Intl1	56 °C	IntI1F (ACGAGCGCAAGGTTTCGGT) IntI1R (GAAAGGTCTGGTCATACATG)	565
Class 2 integrase gene	Intl2	52 °C	IntI2F (GTGCAACGCATTTTGCAGG) IntI2R (CAACGGAGTCATGCAGATG)	403
Class 3 integrase gene	Intl3	57 °C	IntI3F (CATTTGTGTTGTGGACGGC) IntI3R (GACAGATACGTGTTTGGCAA)	717
Variable regions		52 °C	5′-CS (GGCATCCAAGCAGCAAG) 3′-CS (AAGCAGACTTGACCTGAT)	Uncertain

**Table 5 microorganisms-11-02411-t005:** Antimicrobial susceptibility profile of the XDR (*n* = 21) and PDR (*n* = 2) isolates of *K. pneumoniae*.

Isolate	AMP	CTX	FEP	CIP	CAZ	TZP	FOX	IPM	MEM	CN	AK	CL
Kp 1	R	R	R	R	R	R	R	R	R	R	I	S
Kp 2	R	R	R	R	R	R	R	R	R	S	I	R
Kp 5	R	R	R	R	R	R	R	R	R	R	R	S
Kp 6 PDR	R	R	R	R	R	R	R	R	R	R	R	R
Kp 7	R	R	R	R	R	I	R	S	S	R	R	S
Kp 11 PDR	R	R	R	R	R	R	R	R	R	R	R	R
Kp 13	R	R	R	R	R	R	R	I	R	R	R	S
Kp 15	R	R	R	R	R	R	R	R	R	R	R	S
Kp 16	R	R	R	R	R	R	R	R	R	S	S	S
Kp 22	R	R	R	R	R	S	R	R	R	S	S	R
Kp 24	R	R	R	R	R	R	R	R	R	S	S	S
Kp 28	R	R	R	R	R	R	R	R	R	R	R	S
Kp 29	R	R	R	R	R	R	R	R	R	R	R	S
Kp 30	R	R	R	R	R	R	R	R	R	R	R	S
Kp 31	R	R	R	R	R	R	R	R	R	R	R	S
Kp 32	R	R	R	R	I	R	R	R	R	R	R	S
Kp 41	R	R	R	R	R	R	R	R	R	R	S	S
Kp 43	R	R	R	R	R	R	R	R	R	R	S	S
Kp 44	R	R	R	R	R	R	R	R	R	R	I	S
Kp 45	R	R	R	R	R	R	R	R	R	R	R	S
Kp 48	R	R	R	R	R	R	R	R	R	R	R	S
Kp 49	R	R	R	R	R	R	R	R	R	R	R	S
Kp 50	R	R	R	R	R	R	R	R	R	R	S	R

Abbreviations: Kp = *Klebsiella pneumoniae*, AMP = ampicillin, CTX = ceftriaxone, FEP = cefepime, CIP = ciprofloxacin, CAZ = ceftazidime, TZP = piperacillin–tazobactam, FOX = cefoxitin, IPM = imipenem, MEM = meropenem, CN = gentamicin, AK = amikacin, CL = colistin.

**Table 6 microorganisms-11-02411-t006:** Demographic features and clinical characteristics of the patients.

Demographic and Clinical Characteristics	Case Patients, *n* = 13
Male, *n* (%)	12 (92%)
Age	
Adults ≤ 24 years, *n* (%)	1 (7.7%)
From 25–50 years, *n* (%)	3 (23.1%)
More than 50 years, *n* (%)	9 (69.2%)
Age at first positive culture	
Mean age of adults, years (range)	59.7 (20–86)
Length of stay	
Median length of stay after a first positive culture, days (range)	48 (1–134)
Hospital location	
Intensive Care Unit (ICU), *n* (%)	8 (61.5%)
Intermediate care ward, *n* (%)	5 (83.5%)

**Table 7 microorganisms-11-02411-t007:** The detected amino acid changes in PhoP and PhoQ in colistin-resistant strains (*n* = 4).

Isolate (Kp)	PhoP	PhoQ
Kp 6 and Kp 11	3 SNPs	−Ve
	(Gln147His)	
	(Gln131Glu)	
	(Pro129Leu)	
Kp 22 and Kp 50	3 SNPs	−Ve
	(Val130Glu) (Gln147His)	
	(Gln131Glu)	

**Table 8 microorganisms-11-02411-t008:** Level of agreement of antimicrobial-resistance genes and phenotypic resistance for *K. pneumoniae* isolates.

*K. pneumoniae* (*n* = 23)		Phenotypic Resistance	Positive Genes	Level of Agreement Genotype with Phenotypic Expression %
Aminoglycosides	Amikacin	9	21	42.90%
	Gentamicin	15	21	71.40%
Quinolones		23	23	100%
Carbapenems		14	14	100%
Cephalosporins		23	23	100%

**Table 9 microorganisms-11-02411-t009:** The detected gene cassettes in the *K. pneumoniae* isolates (*n* = 23).

Isolate	ST Type	Gene Cassette
Kp 40	37	*dfrA12*, *ant1*
Kp 49	147	*Arr3*, *ereA2*, *aadA*, *cmlA1*
Kp 21	45	*dfra12*, *ant1*
Kp 41	395	*dfrA14,*
Kp 43	395	*dfrA12*, *APH(3″)-Ia*
Kp 44	395	*dfrA14*
Kp 50	395	*dfrA14*
Kp 16	395	*In0*
Kp 22	395	*In0*
Kp 5	231	*aac(6′)-Ib*, *arr2*
Kp 6	231	*dfrA12*, *emrE*, *ant1*
Kp 7	231	*dfrA12*, *emrE*, *ant1*
Kp 10	231	*dfrA5*
Kp 11	231	*aacA4*, *cat1*, *ant1*
Kp 15	231	*aacA4*, *emrE*, *ant1*
Kp 28	231	*Ant1*, *erm*, *cat1*
Kp 30	231	*aacA4*
Kp 45	231	*dfrA14*
Kp 42	405	*dfrA14*
Kp 46	405	*dfrA14*
Kp 27	280	*dfrA14*
Kp 25	1741	*In0*
Kp 37	1710	*dfrA14*

**Table 10 microorganisms-11-02411-t010:** Plasmids existent in *K. pneumoniae* isolates and their features.

Isolate (KP)	Plasmid	Size (bp)	Replicon	Resistance Genes
5, 6, 7, 10, 11, 15, 16, 22, 28, 30, 41, 43, 44, 45, 50	pKPQIL-IT	115,300	IncFIB (QIL)	*bla_TEM-1_*, *bla_KPC-3_*
5, 6, 11, 15, 16, 22, 28, 30, 41, 43, 44, 45, 49, 50	pKP3-A	7605	ColKP3	*Bla_OXA-181_*
5, 6, 7, 15, 28, 30, 45	pAMA1167-NDM-5	11,310	IncFII (pAMA1167-NDM-5)	*aadA5*, *aadA2*, *aac(3)-IId*, *aph(6)-Id*, *aph(3″)-Ib, aac(6′)-Ib-cr5, bla*_NDM-5_, *bla*_OXA-1_, *bla*_CTX-M-15-1_, *bla*_TEM-1_ *dfrA17*, *dfrA12, Mph(A)*, *Sul1*, *Sul2*, *emrE*, *tet(b)*, *tet(C)*, *cat*
10, 25, 27, 37, 42, 46	pKPN-IT	208,191	IncFIB (K)	*aadA2*, *cat, Mph(A)*, *Sul1*, *dfrA12*
21, 40	pCAV1099-14	113,992	IncFIB(K) (pCAV1099-14)	*dfrA19*, *APH(3″)-Ib, APH(3′)-Ia*, *QnrB52*
41, 50	pNDM-MAR	267,242	IncFIB (pNDM-Mar) IncHI1B (pNDM-Mar)	*aac(6′)-Ib, bla_OXA-1_, bla_NDM-1_, cat, QnrB1*
27, 49	pK245	98,264	IncR	*aacC2*, *strA*, *strB*, *dfrA14*, *catA2*, *Qnrs*, *bla_SH2A_*, *bla_TEM_*
49	pC15-1a	92,353	IncFII	*aac(6′)-Ib*, *aac(3)-II*, *bla_TEM-1_*, *bla_OXA_*-*_1_*, *bla_CTX-M-15_*-*_1_*, *tet(A)*
37	pBK30683	139,941	FIA (pBK30683)	*dfrA14*, *StrA*, *StrB*, *bla_TEM-1_*, *bla_OXA_*-_9_, *bla_KPC-3_ Sul2,ant(3″)-Ia*

## Data Availability

All supporting data can be found in the Appendix A. All whole-genome sequencing data were deposited in GenBank at accession number PRJNA999478.

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
