# Peer review of "Comparative Genomic Analysis Reveals the Emergence of ST-231 and ST-395 Klebsiella pneumoniae Strains Associated with the High Transmissibility of blaKPC Plasmids"

_microorganisms, 2023, doi:10.3390/microorganisms11102411_

Round 1
Reviewer 1 Report
Comments to the manuscript ID: microorganisms-2618769 “Comparative Genomic Analysis reveals Emergence of ST-231 and ST-395 Klebsiella pneumoniae strains associated with high transmissibility of blaKPC plasmids”
The results and findings described in the manuscript are significant both for the scientific and the general public. The authors did a great job of isolating, characterizing the isolates, typing and sequencing a large number of genomes, while for some strains they also performed conjugation experiments to show the transmissibility of plasmids carrying resistance to antibiotics. Analysis and control of strains/isolates from clinics possessing the ability to transmit is important in order to reduce the spread of resistance of pathogenic strains.
Unfortunately, the manuscript is not very well written, there is a lot of repetition and redundant information, inadequately cited literature, so it requires a serious revision. The discussion is too long with a lot of repeated information so it is necessary to shorten and summarize.
Specific comments
Line 25. BlaOXA-232 and blaCTX-M-15 were the most frequently detected carbapenems and beta-lactamases.
Should be carbapenemases (not carbapenems).
Line 29. ….multidrug-resistant organisms.
Bacteria, strains, pathogens (not organisms; the term "organism" is most often used for higher organisms).
Lines 34-36. Klebsiella pneumoniae is an opportunistic pathogen that is associated with different serious nosocomial infections including pneumonia, septicemia, meningitis and urinary tract infections (UTIs) [1,2].
Cited manuscripts [1,2] are too narrow/specific regarding author`s statement. Better citations for example manuscripts:
Ashurst JV, Dawson A. Klebsiella Pneumonia. [Updated 2023 Jul 20]. In: StatPearls [Internet]. Treasure Island (FL): StatPearls Publishing; 2023 Jan-. Available from: https://www.ncbi.nlm.nih.gov/books/NBK519004
Podschun R, Ullmann U. Klebsiella spp. as nosocomial pathogens: epidemiology, taxonomy, typing methods, and pathogenicity factors. Clin Microbiol Rev. 1998 Oct;11(4):589-603. doi: 10.1128/CMR.11.4.589. PMID: 9767057; PMCID: PMC88898.
Line 38. ……however they are considered as last resort antibiotics [3].
Also, for statement that Carbapenems are considered as last-resort antibiotics for the treatment of infections caused by multidrug-resistant Gram-negative bacteria better citation is:
Aurilio C, Sansone P, Barbarisi M, Pota V, Giaccari LG, Coppolino F, Barbarisi A, Passavanti MB, Pace MC. Mechanisms of Action of Carbapenem Resistance. Antibiotics. 2022; 11(3):421. https://doi.org/10.3390/antibiotics11030421
Lines 38-39. Moreover, the number of K. pneu-38 moniae carbapenemase (KPC) enzyme producers has been increasing [4,5].
Better citations are for this statement:
Arnold RS, Thom KA, Sharma S, Phillips M, Kristie Johnson J, Morgan DJ. Emergence of Klebsiella pneumoniae carbapenemase-producing bacteria. South Med J. 2011 Jan;104(1):40-5. doi: 10.1097/SMJ.0b013e3181fd7d5a. PMID: 21119555; PMCID: PMC3075864.
Aires-de-Sousa M, Ortiz de la Rosa JM, Gonçalves ML, Pereira AL, Nordmann P, Poirel L. Epidemiology of Carbapenemase-Producing Klebsiella pneumoniae in a Hospital, Portugal. Emerg Infect Dis. 2019 Sep;25(9):1632-1638. doi: 10.3201/eid2509.190656. PMID: 31441424; PMCID: PMC6711212.
Ainoda, Y., Aoki, K., Ishii, Y. et al. Klebsiella pneumoniae carbapenemase (KPC)-producing Klebsiella pneumoniaeST258 isolated from a Japanese patient without a history of foreign travel - a new public health concern in Japan: a case report. BMC Infect Dis 19, 20 (2019). https://doi.org/10.1186/s12879-018-3649-9
https://www.emro.who.int/emhj-vol-19-2013/11/high-prevalence-of-klebsiella-pneumoniae-carbapenemase-mediated-resistance-in-k-pneumoniae-isolates-from-egypt.html
Lines 39-41. The spread of resistance determinants have been facilitated by horizontal gene transfer mechanisms mainly via mobile genetic elements (MGEs).
Please add reference for this statement like:
Tao S, Chen H, Li N, Wang T, Liang W. The Spread of Antibiotic Resistance Genes In Vivo Model. Can J Infect Dis Med Microbiol. 2022 Jul 18;2022:3348695. doi: 10.1155/2022/3348695. PMID: 35898691; PMCID: PMC9314185.
Lines 47-49. A significant association between integron positive isolates and antibiotic resistance for some drugs were observed……..
References like
Derakhshan S, Najar Peerayeh S, Bakhshi B. Association Between Presence of Virulence Genes and Antibiotic Resistance in Clinical Klebsiella Pneumoniae Isolates. Lab Med. 2016 Nov;47(4):306-311. doi: 10.1093/labmed/lmw030. Epub 2016 Aug 7. PMID: 27498999.
Firoozeh, F., Mahluji, Z., Khorshidi, A. et al. Molecular characterization of class 1, 2 and 3 integrons in clinical multi-drug resistant Klebsiella pneumoniae isolates. Antimicrob Resist Infect Control 8, 59 (2019). https://doi.org/10.1186/s13756-019-0509-3
Jahanbin F, Marashifard M, Jamshidi S, Zamanzadeh M, Dehshiri M, Malek Hosseini SAA, Khoramrooz SS. Investigation of Integron-Associated Resistance Gene Cassettes in Urinary Isolates of Klebsiella pneumoniae in Yasuj, Southwestern Iran During 2015-16. Avicenna J Med Biotechnol. 2020 Apr-Jun;12(2):124-131. PMID: 32431797; PMCID: PMC7229451.
are more adequate.
Lines 49-51. Both classes 1 and class 2 integrons have the gene cassettes encoding resistance to trimethoprim (dfr) as a predominant gene, which may be due to long-term usage of this antibiotic [14].
More adequate reference for this statement are:
Deng Y, Bao X, Ji L, Chen L, Liu J, Miao J, Chen D, Bian H, Li Y, Yu G. Resistance integrons: class 1, 2 and 3 integrons. Ann Clin Microbiol Antimicrob. 2015 Oct 20;14:45. doi: 10.1186/s12941-015-0100-6. PMID: 26487554; PMCID: PMC4618277.
Solberg OD, Ajiboye RM, Riley LW. Origin of class 1 and 2 integrons and gene cassettes in a population-based sample of uropathogenic Escherichia coli. J Clin Microbiol. 2006 Apr;44(4):1347-51. doi: 10.1128/JCM.44.4.1347-1351.2006. PMID: 16597861; PMCID: PMC1448660.
Line 59. However, molecular charecterization…..
Characterization not charecterization ……
Lines 62-63. In addition, we explored the association between antimicrobial susceptibility and integron carriage in clinical isolates of K. pneumoniae
In addition, we investigated the association between antimicrobial susceptibility and the presence of integron(s) in clinical isolates of K. pneumoniae.
Lines 72-73. The isolates were mostly from urine (n= 25), respiratory (n= 10), wound (n= 9), bloodstream (n= 4), body fluid (n=1) and biopsies (n=1) samples.
Line 78. Table 1. Clinical information of K. pneumoniae isolates (n=50).
In Table 1. Since all isolates are from 2019 it is better to specify month of isolation.
Lines 91-94. Concentrations of antibiotics on disks are in mg (milligrams) should be in µg (micrograms).
Also, in Table 2.
Lines 113-114. Then, bacterial suspension was centrifuged the next day for 15 minutes at 4000 x g.
Delete “next day” because it is not important and it is logical that it is the next day
Lines 114-117. The amount of 20-40mg of pelleted bacterial cells was re-suspended in a previously prepared 100 μL of 0.1mg/mL pre-lysis buffer (100 μl TE buffer and 0.1 μl RNase A 100mg/mL) (Thermo Fisher Scientific, Winsford, UK
This sentence is not clear. How it is possible that pre-lysis buffer could be 0.1 mg/ml
I know that it refers to the concentration of RNase, but it is not well written, please to paraphrase.
Line 117. All cells were well resuspended by pipetting up and down several times.
Please delete “All” because how you can confirm that “all” cells are resuspended??)
Line 118. After that, the sample was incubated again at 37°C at a 400 rpm shaking incubator for……
“samples were” because authors analysed 50 isolates???
Line 123. The pellet was re-suspended in 50-100 µl….
Which pellet? DNA pellet?
I don't understand how someone can jump from a pellet of bacteria to a pellet of DNA without any explanation of what was done in between?
Lines 125-126. The boiling method was used to extract the DNA from transconjugant colonies in the conjugation experiments.
Please add reference for boiling method
for example
Queipo-Ortuño MI, De Dios Colmenero J, Macias M, Bravo MJ, Morata P. Preparation of bacterial DNA template by boiling and effect of immunoglobulin G as an inhibitor in real-time PCR for serum samples from patients with brucellosis. Clin Vaccine Immunol. 2008 Feb;15(2):293-6. doi: 10.1128/CVI.00270-07. Epub 2007 Dec 12. PMID: 18077622; PMCID: PMC2238042.
Lines 136-139. The PCR reaction mixture consists of 5ml of 5X PCR green buffer; 0.5 ml of dNTPs (10mM); 1ml of both forward and reverse primers (10 mM); 0.1ml of Go-Tag DNA polymerase (5m/ml); and 1.5ml of template DNA. Then the nuclease-free water was added until the total volume reached 25 ml.
PCR reaction mixture volumes are probably done in 1000 times smaller volumes µl (not ml).
Lines 147-148. The specific primers for detecting integrase genes, variable regions, and capsules as previously described [14].
I would like to at least state for the first time that these are integron "variable regions" since there are other variable regions in genomes
Table 3. “Initialization” should be “Initial denaturation“, like “Final extension” not “Finalization”
Lines 151-163. 2.4.1. PCR purification
Why here is complete procedure for PCR product purification was written in detail, while other are just indicated as "according manufacturer`s instructions"???
Lines 166-171. 2.5. Gel electrophoresis
All PCR products were visualized by using 2% agarose gel electrophoresis containing 167 1x TBE (ThermoFisher Scientific, USA) and MIDORIGreen Direct (NIPPON-genetics, Eu-168 rope).
Gel electrophoresis is already mentioned/explained in previous paragraph, please delete one.
Lines 175-177. The genomic DNA was extracted and prepared and then was sent for sequencing following the protocol provided by the sequencing facility.
and, and, and
Please delete some "and"
Line 181. epidemiology server (CGE) (http://cge.cbs.dtu.dk/services, accessed on 02nd July 2020).
How could the authors access CGE on July 2, 2020, when they received the data on July 20, 2022, almost two years later?
Line 205. …regions as donors, and E. coli HB101 as a recipient by using filter mating method [27].
Sorry in reference 27.
27. Xu, H.; Davies, J.; Miao, V. Molecular Characterization of Class 3 Integrons from Delftia Spp. J Bacteriol 2007, 189 (17). 807 https://doi.org/10.1128/JB.00348-07. 808
I cannot find filter mating method
I found similar to this mating experiments for example in manuscript
Velhner M, Todorović D, Novović K, Jovčić B, Lazić G, Kojić M, Kehrenberg C. Characterization of antibiotic resistance in Escherichia coli isolates from Black-headed gulls (Larus ridibundus) present in the city of Novi Sad, Serbia. Vet Res Commun. 2021 Dec;45(4):199-209. doi: 10.1007/s11259-021-09801-7. Epub 2021 Jun 18. PMID: 34142260.
Lines 237-238. (Gel electrophoresis is presented in supplementary files).
In addition, This sentence should immediately after Table 4. because there is no electrophoresis without amplified fragments for integron class 2 and 3.
Line 242. Moreover, 88% of the isolates showed resistance to ciprofloxacin (88%).
Two times written (88%), delete one please.
Lines 242-244. Furthermore, these isolates showed intermediate resistance to gentamicin (54%), imipenem (52%), and a lesser extent to amikacin and colistin (30% and 6%), respectively.
Please check if it is intermediate resistance or about 50% of isolates since intermediate resistance is presented by dark colour (for gentamicin 0, imipenem 2%, amikacin 8%, colistin 0, ???
Percentages present in text are for resistant isolates not for intermediate (Please see Figure1).
Lines 248-249. According to the results obtained by phoenixBD semi-automated system, 54% of K. pneumoniae isolates (n=27) were identified as ESBLs, whereas 42 % (n=21)….
Line 250. PDR. All XDRs were susceptible to colistin except 3…
Almost all XDR isolates were susceptible to colistin except 3….
Line 261. Seven representative samples were selected to be…
Not samples (isolates/strains…)! Also, on other places.
Line 267. .. detected in all trans-conjugants which conforms that these conjugative plasmids are
Please indicate that E. coli transconjugants were obtained (please point out it “E. coli”).
Line 278. …..and 2 isolates showed bands with unexpected size for intI3 gene (Kp 4 and KP 22).
According to Filipic et al. 2023 there are variants of integrese gene/protein.
Filipić B, Malešević M, Vasiljević Z, Novović K, Kojić M, Jovčić B. Comparative genomics of trimethoprim-sulfamethoxazole-resistant Achromobacter xylosoxidans clinical isolates from Serbia reveals shortened variant of class 1 integron integrase gene. Folia Microbiol (Praha). 2023 Jun;68(3):431-440. doi: 10.1007/s12223-022-01026-8. Epub 2022 Dec 26. PMID: 36567375.
Line 280. However, WGS showed that two of these isolates were positive for class 1 integron…..
Please explain why amplification was not observed by PCR (in discussion).
Line 316. The demographic data for thirteen samples that belong to ST-231 (n=9) and ST-395
Again samples; should be strains/ isolates.
Also, line 338, line 405
Line 364. …follows: (Val129Glu), (Gln147His), (Gln131Glu), and (Pro129Thr) (Table 7.).
I do not understand how can be two substitutions at position 129 of PhoP gene/protein (Val129Glu) and (Pro129Thr)
According to UniProt PhoP sequence at position 129 is Leu
Please be more precise with comparing and analysis of SNP
Line 419. In this isolate, blaNDM-1 was within pNDM-Mar plasmid.
The previous sentence listed two isolates, which one????
Table 11. Size (bp) of the plasmids is divided into two lines and creates confusion!
In addition, there is no Table 10?????
Line 460. The absence of both class 2 and class 3 integrons in our isolates is expected since these………
What means "our isolates"? please explain.
Also, Line 524,
Lines 470-471. ….amikacin and gentamicin were resistant to 51.6% and 45.2%, respectively (Derakhshan et al., 2013).
Derakhshan et al. 2013 is not present in Reference list. Please add.
In addition, reference should be presented in text as number.
Line 491. …classical structure of integrons where intI1 and/or 5’ end of integrons is truncated were..
It was shown for Achromobacter xylosoxidans by Filipic et al. 2023.
Filipić B, Malešević M, Vasiljević Z, Novović K, Kojić M, Jovčić B. Comparative genomics of trimethoprim-sulfamethoxazole-resistant Achromobacter xylosoxidans clinical isolates from Serbia reveals shortened variant of class 1 integron integrase gene. Folia Microbiol (Praha). 2023 Jun;68(3):431-440. doi: 10.1007/s12223-022-01026-8. Epub 2022 Dec 26. PMID: 36567375.
Discussion too long, Conclusions should be stronger.
should be improved
Author Response
I would like to express my sincere thanks to our reviewer who took the responsibility to thoroughly read the manuscript. I assure you that all suggestions have been taken into considerations. We have now updated all our references according to the recommendations from the reviewer.
In addition, we implemented all the paraphrasing of the sentences to be adjusted.
The discussion section was shortened to be concise and on point.
We addressed all suggestions point by point as follows:
Line 25. BlaOXA-232 and blaCTX-M-15 were the most frequently detected carbapenems and beta-lactamases.
Should be carbapenemases (not carbapenems). Done
Line 29. ….multidrug-resistant organisms.
Bacteria, strains, pathogens (not organisms; the term "organism" is most often used for higher organisms). Done
Lines 34-36. Klebsiella pneumoniae is an opportunistic pathogen that is associated with different serious nosocomial infections including pneumonia, septicemia, meningitis and urinary tract infections (UTIs) [1,2].
Cited manuscripts [1,2] are too narrow/specific regarding author`s statement. Better citations for example manuscripts:
Ashurst JV, Dawson A. Klebsiella Pneumonia. [Updated 2023 Jul 20]. In: StatPearls [Internet]. Treasure Island (FL): StatPearls Publishing; 2023 Jan-. Available from: https://www.ncbi.nlm.nih.gov/books/NBK519004
Podschun R, Ullmann U. Klebsiella spp. as nosocomial pathogens: epidemiology, taxonomy, typing methods, and pathogenicity factors. Clin Microbiol Rev. 1998 Oct;11(4):589-603. doi: 10.1128/CMR.11.4.589. PMID: 9767057; PMCID: PMC88898. Done
Line 38. ……however they are considered as last resort antibiotics [3].
Also, for statement that Carbapenems are considered as last-resort antibiotics for the treatment of infections caused by multidrug-resistant Gram-negative bacteria better citation is:
Aurilio C, Sansone P, Barbarisi M, Pota V, Giaccari LG, Coppolino F, Barbarisi A, Passavanti MB, Pace MC. Mechanisms of Action of Carbapenem Resistance. Antibiotics. 2022; 11(3):421. https://doi.org/10.3390/antibiotics11030421. Done
Lines 38-39. Moreover, the number of K. pneu-38 moniae carbapenemase (KPC) enzyme producers has been increasing [4,5].
Better citations are for this statement: All four citations done
Arnold RS, Thom KA, Sharma S, Phillips M, Kristie Johnson J, Morgan DJ. Emergence of Klebsiella pneumoniae carbapenemase-producing bacteria. South Med J. 2011 Jan;104(1):40-5. doi: 10.1097/SMJ.0b013e3181fd7d5a. PMID: 21119555; PMCID: PMC3075864.
Aires-de-Sousa M, Ortiz de la Rosa JM, Gonçalves ML, Pereira AL, Nordmann P, Poirel L. Epidemiology of Carbapenemase-Producing Klebsiella pneumoniae in a Hospital, Portugal. Emerg Infect Dis. 2019 Sep;25(9):1632-1638. doi: 10.3201/eid2509.190656. PMID: 31441424; PMCID: PMC6711212.
Ainoda, Y., Aoki, K., Ishii, Y. et al. Klebsiella pneumoniae carbapenemase (KPC)-producing Klebsiella pneumoniaeST258 isolated from a Japanese patient without a history of foreign travel - a new public health concern in Japan: a case report. BMC Infect Dis 19, 20 (2019). https://doi.org/10.1186/s12879-018-3649-9
https://www.emro.who.int/emhj-vol-19-2013/11/high-prevalence-of-klebsiella-pneumoniae-carbapenemase-mediated-resistance-in-k-pneumoniae-isolates-from-egypt.html
Lines 39-41. The spread of resistance determinants have been facilitated by horizontal gene transfer mechanisms mainly via mobile genetic elements (MGEs).
Please add reference for this statement like:
Tao S, Chen H, Li N, Wang T, Liang W. The Spread of Antibiotic Resistance Genes In Vivo Model. Can J Infect Dis Med Microbiol. 2022 Jul 18;2022:3348695. doi: 10.1155/2022/3348695. PMID: 35898691; PMCID: PMC9314185.
Lines 47-49. A significant association between integron positive isolates and antibiotic resistance for some drugs were observed……..
References like All three citations done
Derakhshan S, Najar Peerayeh S, Bakhshi B. Association Between Presence of Virulence Genes and Antibiotic Resistance in Clinical Klebsiella Pneumoniae Isolates. Lab Med. 2016 Nov;47(4):306-311. doi: 10.1093/labmed/lmw030. Epub 2016 Aug 7. PMID: 27498999.
Firoozeh, F., Mahluji, Z., Khorshidi, A. et al. Molecular characterization of class 1, 2 and 3 integrons in clinical multi-drug resistant Klebsiella pneumoniae isolates. Antimicrob Resist Infect Control 8, 59 (2019). https://doi.org/10.1186/s13756-019-0509-3
Jahanbin F, Marashifard M, Jamshidi S, Zamanzadeh M, Dehshiri M, Malek Hosseini SAA, Khoramrooz SS. Investigation of Integron-Associated Resistance Gene Cassettes in Urinary Isolates of Klebsiella pneumoniae in Yasuj, Southwestern Iran During 2015-16. Avicenna J Med Biotechnol. 2020 Apr-Jun;12(2):124-131. PMID: 32431797; PMCID: PMC7229451.
are more adequate.
Lines 49-51. Both classes 1 and class 2 integrons have the gene cassettes encoding resistance to trimethoprim (dfr) as a predominant gene, which may be due to long-term usage of this antibiotic [14].
More adequate reference for this statement are:
Deng Y, Bao X, Ji L, Chen L, Liu J, Miao J, Chen D, Bian H, Li Y, Yu G. Resistance integrons: class 1, 2 and 3 integrons. Ann Clin Microbiol Antimicrob. 2015 Oct 20;14:45. doi: 10.1186/s12941-015-0100-6. PMID: 26487554; PMCID: PMC4618277.
Solberg OD, Ajiboye RM, Riley LW. Origin of class 1 and 2 integrons and gene cassettes in a population-based sample of uropathogenic Escherichia coli. J Clin Microbiol. 2006 Apr;44(4):1347-51. doi: 10.1128/JCM.44.4.1347-1351.2006. PMID: 16597861; PMCID: PMC1448660. All three citations done
Line 59. However, molecular charecterization….. done
Characterization not charecterization …… done
Lines 62-63. In addition, we explored the association between antimicrobial susceptibility and integron carriage in clinical isolates of K. pneumoniae
In addition, we investigated the association between antimicrobial susceptibility and the presence of integron(s) in clinical isolates of K. pneumoniae. done
Lines 72-73. The isolates were mostly from urine (n= 25), respiratory (n= 10), wound (n= 9), bloodstream (n= 4), body fluid (n=1) and biopsies (n=1) samples. done
Line 78. Table 1. Clinical information of K. pneumoniae isolates (n=50). done
In Table 1. Since all isolates are from 2019 it is better to specify month of isolation. Changed to months done
Lines 91-94. Concentrations of antibiotics on disks are in mg (milligrams) should be in µg (micrograms). done
Also, in Table 2. done
Lines 113-114. Then, bacterial suspension was centrifuged the next day for 15 minutes at 4000 x g.
Delete “next day” because it is not important and it is logical that it is the next day done
Lines 114-117. The amount of 20-40mg of pelleted bacterial cells was re-suspended in a previously prepared 100 μL of 0.1mg/mL pre-lysis buffer (100 μl TE buffer and 0.1 μl RNase A 100mg/mL) (Thermo Fisher Scientific, Winsford, UK
This sentence is not clear. How it is possible that pre-lysis buffer could be 0.1 mg/ml
I know that it refers to the concentration of RNase, but it is not well written, please to paraphrase. Sentence changed
Line 117. All cells were well resuspended by pipetting up and down several times.
Please delete “All” because how you can confirm that “all” cells are resuspended??) Sentence changed
Line 118. After that, the sample was incubated again at 37°C at a 400 rpm shaking incubator for……
“samples were” because authors analysed 50 isolates??? done
Line 123. The pellet was re-suspended in 50-100 µl….
Which pellet? DNA pellet?
I don't understand how someone can jump from a pellet of bacteria to a pellet of DNA without any explanation of what was done in between? The sentences rephrased
Lines 125-126. The boiling method was used to extract the DNA from transconjugant colonies in the conjugation experiments.Please add reference for boiling method for example
Queipo-Ortuño MI, De Dios Colmenero J, Macias M, Bravo MJ, Morata P. Preparation of bacterial DNA template by boiling and effect of immunoglobulin G as an inhibitor in real-time PCR for serum samples from patients with brucellosis. Clin Vaccine Immunol. 2008 Feb;15(2):293-6. doi: 10.1128/CVI.00270-07. Epub 2007 Dec 12. PMID: 18077622; PMCID: PMC2238042. Done
Lines 136-139. The PCR reaction mixture consists of 5ml of 5X PCR green buffer; 0.5 ml of dNTPs (10mM); 1ml of both forward and reverse primers (10 mM); 0.1ml of Go-Tag DNA polymerase (5m/ml); and 1.5ml of template DNA. Then the nuclease-free water was added until the total volume reached 25 ml.
PCR reaction mixture volumes are probably done in 1000 times smaller volumes µl (not ml). conversion error from the word file
Lines 147-148. The specific primers for detecting integrase genes, variable regions, and capsules as previously described [14].
I would like to at least state for the first time that these are integron "variable regions" since there are other variable regions in genomes Done
Table 3. “Initialization” should be “Initial denaturation“, like “Final extension” not “Finalization” Done
Lines 151-163. 2.4.1. PCR purification
Why here is complete procedure for PCR product purification was written in detail, while other are just indicated as "according manufacturer`s instructions"??? it was detailed because there were some modifications from the original protocol.
Lines 166-171. 2.5. Gel electrophoresis
All PCR products were visualized by using 2% agarose gel electrophoresis containing 167 1x TBE (ThermoFisher Scientific, USA) and MIDORIGreen Direct (NIPPON-genetics, Eu-168 rope).
Gel electrophoresis is already mentioned/explained in previous paragraph, please delete one. Done
Lines 175-177. The genomic DNA was extracted and prepared and then was sent for sequencing following the protocol provided by the sequencing facility.
and, and, and
Please delete some "and" Done
Line 181. epidemiology server (CGE) (http://cge.cbs.dtu.dk/services, accessed on 02nd July 2020).
How could the authors access CGE on July 2, 2020, when they received the data on July 20, 2022, almost two years later? Error modified
Line 205. …regions as donors, and E. coli HB101 as a recipient by using filter mating method [27].
Sorry in reference 27.
- Xu, H.; Davies, J.; Miao, V. Molecular Characterization of Class 3 Integrons from Delftia Spp. J Bacteriol 2007, 189 (17). 807 https://doi.org/10.1128/JB.00348-07. 808
I cannot find filter mating method
I found similar to this mating experiments for example in manuscript
Velhner M, Todorović D, Novović K, Jovčić B, Lazić G, Kojić M, Kehrenberg C. Characterization of antibiotic resistance in Escherichia coli isolates from Black-headed gulls (Larus ridibundus) present in the city of Novi Sad, Serbia. Vet Res Commun. 2021 Dec;45(4):199-209. doi: 10.1007/s11259-021-09801-7. Epub 2021 Jun 18. PMID: 34142260. Done
Lines 237-238. (Gel electrophoresis is presented in supplementary files).
In addition, This sentence should immediately after Table 4. because there is no electrophoresis without amplified fragments for integron class 2 and 3. Moved after table 4 in methods
Line 242. Moreover, 88% of the isolates showed resistance to ciprofloxacin (88%).
Two times written (88%), delete one please. Done
Lines 242-244. Furthermore, these isolates showed intermediate resistance to gentamicin (54%), imipenem (52%), and a lesser extent to amikacin and colistin (30% and 6%), respectively.
Please check if it is intermediate resistance or about 50% of isolates since intermediate resistance is presented by dark colour (for gentamicin 0, imipenem 2%, amikacin 8%, colistin 0, ???
Percentages present in text are for resistant isolates not for intermediate (Please see Figure1). Yes we confirm that the Percentages present in text are for resistant isolates not for intermediate
Lines 248-249. According to the results obtained by phoenixBD semi-automated system, 54% of K. pneumoniae isolates (n=27) were identified as ESBLs, whereas 42 % (n=21)….done
Line 250. PDR. All XDRs were susceptible to colistin except 3…
Almost all XDR isolates were susceptible to colistin except 3…..done
Line 261. Seven representative samples were selected to be…
Not samples (isolates/strains…)! Also, on other places. .done
Line 267. .. detected in all trans-conjugants which conforms that these conjugative plasmids are
Please indicate that E. coli transconjugants were obtained (please point out it “E. coli”). done
Line 278. …..and 2 isolates showed bands with unexpected size for intI3 gene (Kp 4 and KP 22).
According to Filipic et al. 2023 there are variants of integrese gene/protein. done
Filipić B, Malešević M, Vasiljević Z, Novović K, Kojić M, Jovčić B. Comparative genomics of trimethoprim-sulfamethoxazole-resistant Achromobacter xylosoxidans clinical isolates from Serbia reveals shortened variant of class 1 integron integrase gene. Folia Microbiol (Praha). 2023 Jun;68(3):431-440. doi: 10.1007/s12223-022-01026-8. Epub 2022 Dec 26. PMID: 36567375.
Line 280. However, WGS showed that two of these isolates were positive for class 1 integron…..
Please explain why amplification was not observed by PCR (in discussion). done
Line 316. The demographic data for thirteen samples that belong to ST-231 (n=9) and ST-395
Again samples; should be strains/ isolates. Done in the entire manuscript where applicable
Also, line 338, line 405
Line 364. …follows: (Val129Glu), (Gln147His), (Gln131Glu), and (Pro129Thr) (Table 7.).
I do not understand how can be two substitutions at position 129 of PhoP gene/protein (Val129Glu) and (Pro129Thr)
According to UniProt PhoP sequence at position 129 is Leu
Please be more precise with comparing and analysis of SNP done
Line 419. In this isolate, blaNDM-1 was within pNDM-Mar plasmid.
The previous sentence listed two isolates, which one???? Revised and modified
Table 11. Size (bp) of the plasmids is divided into two lines and creates confusion!
In addition, there is no Table 10????? Error addressed. No table 10 (table 11 is now table 10 and the format adjusted to fit the plasmid size in one line
Line 460. The absence of both class 2 and class 3 integrons in our isolates is expected since these………
What means "our isolates"? please explain. Rephrased
Also, Line 524, all adjusted as the isolates in this study
Lines 470-471. ….amikacin and gentamicin were resistant to 51.6% and 45.2%, respectively (Derakhshan et al., 2013).
Derakhshan et al. 2013 is not present in Reference list. Please add. added
In addition, reference should be presented in text as number.adjusted
Line 491. …classical structure of integrons where intI1 and/or 5’ end of integrons is truncated were..
It was shown for Achromobacter xylosoxidans by Filipic et al. 2023.
Filipić B, Malešević M, Vasiljević Z, Novović K, Kojić M, Jovčić B. Comparative genomics of trimethoprim-sulfamethoxazole-resistant Achromobacter xylosoxidans clinical isolates from Serbia reveals shortened variant of class 1 integron integrase gene. Folia Microbiol (Praha). 2023 Jun;68(3):431-440. doi: 10.1007/s12223-022-01026-8. Epub 2022 Dec 26. PMID: 36567375
Reviewer 2 Report
The study done by ALMuzahmi et al., involves investigation of the genetic composition of conjugative transposons and phenotypic assessment of multidrug-resistant K. pneumoniae isolates together with horizontal transferability. Finally, whole genome sequencing was performed to determine the sequence type (ST), acquired resistome and plasmidome of integron-carrying strains. The study highlighted the high transmissibility of MDR-conferring conjugative plasmids in clinical isolates of K. pneumoniae. The study reveals a significant occurrence of class 1 integrons (96%) in multidrug-resistant Klebsiella pneumoniae, with the absence of class 2 and class 3 integrons. Whole genome sequencing data confirmed the presence of various gene cassettes carried by integrons, with dfrA being the most prevalent cassette. Additionally, two major sequence types (ST-231 and ST-395) were identified, with OXA-232 carbapenemase being the predominant type, while the NDM-5 variant was found in a single isolate. The authors have successfully presented the technical aspects of the technique in a clear and concise manner, making it accessible to both experts and researchers who are new to this field. The inclusion of illustrative figures and diagrams further enhances the understanding of the complex sequencing processes involved. The discussion is well-written and informative. Thus, I recommend acceptance for publication after minor edits:
1- Section 2.4. Polymerase Chain Reaction (PCR)
Please revise the units of volumes used (µL instead of mL), (µM instead of mM)
2- Section 2.4.1 PCR purification
Also, revise the units (µL instead of mL)
3- In the supplementary figure 1 D
Correct the figure caption D (2-11) instead of (2-10), also Fig 1 C (isolates 12, 13, 14 and 15 are repeated)
4- In the supplementary figure 2 B
Correct the figure caption B (2-19) instead of (2-18)
5- Section 2.8. Conjugation experiment line 220 (200 mL of PBS) (µL instead of mL)
6- Line 296: Replace ST-13, ST-17, and ST-10 with ST-1710
7- Figures numbers need to be corrected and I recommend removing the duplicates (between the main text and the supplementary) from the main text and keeping them in the supplementary files only after correction of the numbering.
Author Response
Reviewer 2
1- Section 2.4. Polymerase Chain Reaction (PCR)
Please revise the units of volumes used (µL instead of mL), (µM instead of mM) done
2- Section 2.4.1 PCR purification
Also, revise the units (µL instead of mL) done
3- In the supplementary figure 1 D
Correct the figure caption D (2-11) instead of (2-10), also Fig 1 C (isolates 12, 13, 14 and 15 are repeated) done
4- In the supplementary figure 2 B
Correct the figure caption B (2-19) instead of (2-18) done
5- Section 2.8. Conjugation experiment line 220 (200 mL of PBS) (µL instead of mL) done
6- Line 296: Replace ST-13, ST-17, and ST-10 with ST-1710 done
7- Figures numbers need to be corrected and I recommend removing the duplicates (between the main text and the supplementary) from the main text and keeping them in the supplementary files only after correction of the numbering.
All figure numbering was revised and duplicated numbers removed. All supplementary figures adjusted.
Round 2
Reviewer 1 Report
The authors significantly improved the quality and precision of the manuscript, making it acceptable for publication, thanks to the acceptance of all the reviewers' suggestions. The previous version of the manuscript also contained significant results, but now it sounds much better.